# Exploring System 1 and 2 communication for latent reasoning in LLMs

## Abstract

Should LLM reasoning live in a separate module, or within a single model's forward pass and representational space? We study dual-architecture latent reasoning, where a fluent Base exchanges latent messages with a Coprocessor, and test two hypotheses aimed at improving *latent communication* over Liu et al. (2024b): (H1) increase channel capacity; (H2) learn communication via joint finetuning. Under matched latent-token budgets on GPT-2 and Qwen-3, H2 is consistently strongest while H1 yields modest gains. A unified soft-embedding baseline—a single model with the same forward pass and shared representations, using the same latent-token budget—nearly matches H2 and surpasses H1, suggesting current dual designs mostly add compute rather than qualitatively improving reasoning. Across GSM8K, ProsQA, and a Countdown stress test with increasing branching factor, scaling the latent-token budget beyond small values fails to improve robustness. Latent analyses show overlapping subspaces with limited specialization, consistent with weak reasoning gains. We conclude dual-model latent reasoning remains promising in principle, but likely requires objectives or training schedules that explicitly shape latent spaces for algorithmic planning.

## 1 Introduction

Large language models (LLMs) trained with web-scale pretraining and alignment have achieved impressive zero-shot reasoning capabilities across diverse tasks (Dubey et al., 2024; Hurst et al., 2024; Yang et al., 2025; Liu et al., 2024a). Despite their strong performance, LLMs are often seen as "fast and fluent" rather than genuinely deliberative—resembling the intuitive, System-1 side of dual-process theories of cognition (Kahneman, 2011). Indeed, recent surveys explicitly describe progress in reasoning LLMs as a shift from System-1-like heuristics to System-2-style deliberation (Li et al., 2025b), highlighting the need for architectures that support more structured reasoning.

The dominant approach today is chain-of-thought (CoT) reasoning (Wei et al., 2022; Liu et al., 2024a), where intermediate steps are verbalized in natural language. While effective, CoT incurs substantial token-level overhead, limits abstraction bandwidth, and constrains inference unnecessarily to the sequential, symbolic space of text (Chen et al., 2025; Qu et al., 2025). As model sizes and context window grow, these inefficiencies become increasingly prohibitive, motivating the search for more compact and expressive reasoning representations.

Latent reasoning (Hao et al., 2024; Liu et al., 2024b; Geiping et al., 2025) offers an alternative that enables the model to perform multi-step inference internally within its continuous hidden states, surfacing only the final answer. This paradigm promises two key advantages. First, reasoning in high-dimensional embeddings provides vastly greater expressive bandwidth than token sequences, potentially allowing richer intermediate computations (Zhang et al., 2025). Second, for combinatorial problems, operating over structured latent abstractions can dramatically reduce the effective search space (Geiping et al., 2025). Such ideas find parallels in cognitive science, where humans are believed to reason in internal "mentalese" before translating thoughts into language (Fodor, 1975).

Most latent-reasoning methods still ask a single network to do both fast association and slow deliberation; e.g., Coconut (Hao et al., 2024) feeds the model's last hidden state back as input, forcing the same space to support next-token prediction and a putative "language of thought," creating a representational tug-of-war. Neuroscience evidence instead points to partially distinct substrates (PFC for deliberative control; striatal circuits for habitual responses (Miller & Cohen, 2001; O'Reilly

& Frank, 2006; Dolan & Dayan, 2013)), suggesting that separating roles could help. KV-cache Coprocessors (Liu et al., 2024b) fit this separation, but reported gains are limited; we argue the bottleneck is *latent communication* between modules.

We therefore revisit the KV-Coprocessor design with two changes aimed at strengthening communication: (i) *frozen-Base cache augmentation*, where the Coprocessor writes cache edits that reach all layers of the Base, and (ii) *co-finetuning*, where the Base and Coprocessor are trained jointly to make the Base "listen" to latent messages. With matched token budgets and latent counts $N_L$, we evaluate two model families on pretraining and reasoning (GSM8K, ProsQA, Countdown) and analyze whether latents specialize. This lets us test whether these architectures deliver genuine "latent reasoning" rather than merely adding compute.

In brief, our variants surpass Liu et al. (2024b) on multiple metrics; yet, relative to a matched single-model baseline, all tested dual-architecture designs (ours and Liu et al. (2024b)) yield only modest improvements and show no systematicity on reasoning tasks. Interpretability points to overlapping latent subspaces; within this framework, scaling the number of latents $N_L$ seems to mostly add compute rather than structured reasoning—unless, as we show in preliminary experiments, objectives or schedules are used to explicitly enforce latent specialization.

## 2 RELATED WORK

**Reasoning in latent space.** Latent-space methods shift multi-step inference from tokenized CoT into a small set of latent variables, decoupling compute from text length. Representative approaches include Coconut, which feeds continuous "thoughts" back into the model (Hao et al., 2024); differentiable KV-cache augmentation, which separates a Base token predictor from a Coprocessor that edits the cache (Liu et al., 2024b); and compressed/implicit CoT that learns dense contemplation tokens or plans without emitting text (Cheng et al., 2024; Kurtz et al., 2024). Recent surveys synthesize this trend and its claimed benefits (Li et al., 2025a; Zhu et al., 2025). Despite clear token-efficiency gains, reported improvements on *reasoning* are mixed, motivating our study of how to train dual-model systems for effective latent communication.

**Communication and coordination between models.** Our focus on "latent communication" between a Base and a Coprocessor connects to two strands. First, teacher–student transfer suggests that sharing an initialization can facilitate implicit trait transfer, even without explicit supervision (Cloud et al., 2025); this supports initializing the Coprocessor from the Base to align internal representations. Second, multi-agent prompting frameworks use inter-model dialogue to improve reliability (Du et al., 2023; Liang et al., 2023), though recent analyses find gains can be brittle or dataset-dependent (Wang et al., 2024). Finally, our unified *soft-embedding* baseline is grounded in continuous-prompt methods that endow a single network with extra latent capacity via trainable prefix/prompt vectors (Lester et al., 2021; Li & Liang, 2021; Goyal et al.), providing a strong, parameter-efficient alternative to dual-model designs.

## 3 METHODS

### 3.1 PROBLEM SETTING

**Setup and notation.** Let $B_\theta$ be a frozen, pretrained LLM (*Base*) with parameters $\theta$. Given a prompt $x$ and target $y$, a forward pass of $B_\theta$ produces per-layer key–value caches $\{(K_\ell(x;\theta), V_\ell(x;\theta))\}_{\ell=1}^L$, abbreviated $KV_\theta(x)$. We define a set of $M$ augmentation sites indices $\mathcal{T} = \{t_1, \ldots, t_M\}$ within the sequence $x$. At each site $t_m$, a Coprocessor $C_\phi$ with parameters $\phi$ reads $KV_\theta(x)$ up to $t_m$ together with $N_L$ learnable soft tokens and emits a latent sequence $Z \in \mathbb{R}^{N_L \times d}$. At decoding time, $Z$ is injected back into $B_\theta$ to predict the subsequent tokens. During training, gradients flow from the next $N_A$ "ahead" tokens (the supervision window) back to the latents.

**Objective.** The training goal is to learn latents that improve conditional likelihood:

$$\max_{\phi \text{ (and possibly } \theta)} \mathbb{E}_{(x,y)\sim\mathcal{D}}\big[ \log p_\theta\big(y \mid x, \text{inject}(Z; KV_\theta(x))\big)\big],$$

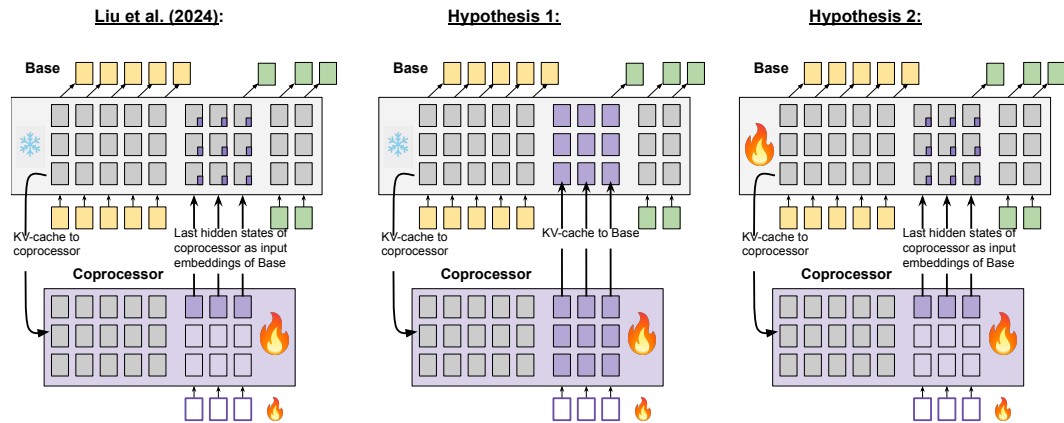

Figure 1: Overview of the deliberation–in–KV-cache architecture of Liu et al. (2024b) and our two variants designed to strengthen cross-module communication.

where $Z = C_\phi(KV_\theta(x), N_L\text{-soft})$ and inject( ) denotes the chosen mechanism for feeding $Z$ (or the Coprocessor's cache) back to the Base. See App B.1 for a more detailed formulation.

**Pipeline (as in Liu et al. (2024b)).** We follow the standard three-stage process (cf. Fig. 1, left):

(i) *KV-cache generation.* Run $B_\theta$ on $x$ once to obtain $KV_\theta(x)$.

(ii) *Latent augmentation.* Run $C_\phi$ on $KV_\theta(x)$ plus $N_L$ learnable soft tokens to produce a latent sequence $Z$. In Liu et al. (2024b), only the final layer's hidden states of $C_\phi$ are used as input embeddings to $B_\theta$.

(iii) *Decoding.* Inject the augmentation ($KV_\theta(x)$ and $Z$) into $B_\theta$ and decode $y$. Unless otherwise noted, only $\phi$ and the soft-token embeddings receive gradients.

### 3.2 EXPERIMENTAL VARIANTS

We test two *hypotheses* aimed at strengthening communication between modules and isolating where the gains arise. Each hypothesis states a falsifiable prediction at matched latent-token budgets.

**Hypothesis 1 — Frozen-Base KV augmentation.** *Change:* Instead of converting the Coprocessor's last layer's output into input embeddings, we concatenate its per-layer cache to the Base cache at injection time. Let $KV_\phi(x)$ denote the Coprocessor's cache produced from $KV_\theta(x)$ and the $N_L$ soft tokens. Decoding uses uses the concatenated caches

$$[K_\ell(x;\theta)\,;\, K_\ell(x;\phi)] \quad \text{and} \quad [V_\ell(x;\theta)\,;\, V_\ell(x;\phi)] \qquad \forall\, \ell = 1, \dots, L,$$

i.e., concatenation along the sequence (cache) dimension.[1] *Optimization:* $\theta$ remains frozen; only $\phi$ and the soft tokens are trained. *Motivation:* With a frozen Base, steering only via input embeddings gives the Coprocessor influence at the first hidden layer. Cache-level augmentation propagates the latent signal through *all* layers, potentially enabling richer, layer-wise "latent communication". *Prediction:* With $\theta$ frozen, cache augmentation will outperform embedding-only feedback Liu et al. (2024b) at matched latent-token budgets $N_L$.

**Hypothesis 2 — Co-finetuned dual-model.** *Change:* Same injection mechanism as the reference system in Liu et al. (2024b) (latents as input embeddings to the Base), but we unfreeze the Base. *Optimization:* Update both $\theta$ and $\phi$ jointly (plus soft tokens) under the log-likelihood objective above. *Motivation:* Joint training allows $B_\theta$ to learn to "listen" to the Coprocessor's messages rather than treating them as exogenous noise. Although this increases the number of trainable parameters, it isolates whether improved cross-module interaction—rather than mere extra compute—drives downstream gains. *Prediction:* For the same latent injection as Liu et al. (2024b), co-finetuning will

---

[1] We denote sequence-axis concatenation by $[\cdot\,;\,\cdot]$.

outperform the frozen-Base counterpart at matched $N_L$, reflecting learned communication rather than added compute.

**Implementation note.** Both variants are realized within the same three-pass schedule used throughout the paper (§4.1): Base pass to form $KV_\theta(x) \to$ Coprocessor pass to form $Z$ (and, for Hyp. 1, $KV_\phi(x)) \to$ Base decode with injection. This preserves a clean separation between next-token prediction and latent computation.

# 4 EXPERIMENTAL ANALYSIS:

## 4.1 LARGE SCALE DATA TRAINING

We replicate the deliberation KV-cache pipeline of Liu et al. (2024b), but adapt it to a tighter compute budget and to two model families.

**Base models:** GPT-2 (124 M) Radford et al. (2019) and Qwen-3 (0.6 B) Yang et al. (2025) replace the 2-B-parameter Gemma used in the Liu et al. (2024b) paper. This choice allows us to complete all runs within a 3-day window on an $8 \times$ A100 (80 GB) node.

**Token budget:** Liu et al. train for $\approx 2 \times 10^{11}$ tokens (sequence 2048, batch 1024, 100 k steps). We scale this down to 40 B tokens for GPT-2 and 8 B tokens for Qwen-3, which we found sufficient to reproduce their qualitative trends without exceeding our hardware envelope.

**Dataset.** All large-scale training uses *FineWeb-Edu-100BT* Lozhkov et al. (2024).

**Sequence & latent parameters:** We use sequence length $S{=}1024$, latent augmentations per sequence $M{=}64$, ahead tokens $N_A{=}16$ for back-propagating the loss into the Coprocessor (Liu et al. (2024b) use 2048/128/16), and $N_L{=}16$ latents. Figure 2**C/D** show that validation perplexity decreases almost linearly as $N_L$ grows under Hypothesis 2. Training cost scales with the effective per-example context $S + M \cdot (N_L + N_A)$, so larger $N_L$ substantially raises wall-clock. Within our budget, $N_L{=}16$ provided a practical operating point, while Liu et al. (2024b) report their strongest results around $N_L{=}32$ on larger bases. We provide further ablations of these hyper-parameters in App G

**Three-pass training loop implementation (Fig. 7):** Liu et al. (2024b) describe "an efficient training framework . . . in one forward pass", enabled by a custom attention mask that scales to large datasets. Such a mask must support multiple augmentations (M) per sequence; otherwise only one augmentation is trained, yielding two orders of magnitude less signal per token. However, we hypothesize that a single forward pass creates an optimization shortcut: since the Base model is pre-trained and competent while the Coprocessor begins as initialized noise, the joint system is incentivized to ignore the noisy latent channel in favor of the Base's internal representations. This reflects the ideas around path of least resistance in deep learning (Bowman et al., 2015; Geirhos et al., 2020). Furthermore, it is technically unclear how to implement such a bidirectional dependency (Base $\to$ Coprocessor $\to$ Base) within a single forward pass, and Liu et al. (2024b) do not provide implementation details to clarify this. To ensure the Base strictly conditions on the Coprocessor's output and to avoid optimization collapse, we adopt a strict three-pass loop (Figure 7) that architecturally enforces the dependency.

Only the Coprocessor (and, in Hypothesis 2, the Base model) receives gradients. Although this three-pass schedule incurs a 3x wall-clock penalty compared with the unknown "single-pass" implementation, it preserves a clean separation between next-token prediction and latent reasoning—the property we wish to evaluate.

After convergence, models are evaluated greedily on standard pretraining benchmark for small LLMs: HellaSwag, ARC-Easy, SocialIQA, PIQA, and Winogrande.

### 4.1.1 BASELINES

**Baseline 1 - Base model + continued pretraining:** Liu et al. (2024b), compare their dual-model to the initial base checkpoint, meaning the dual system has seen far more data. Instead, we keep the architecture unchanged and simply continue pretraining the base model on the **same number of tokens** used by our dual-model runs.

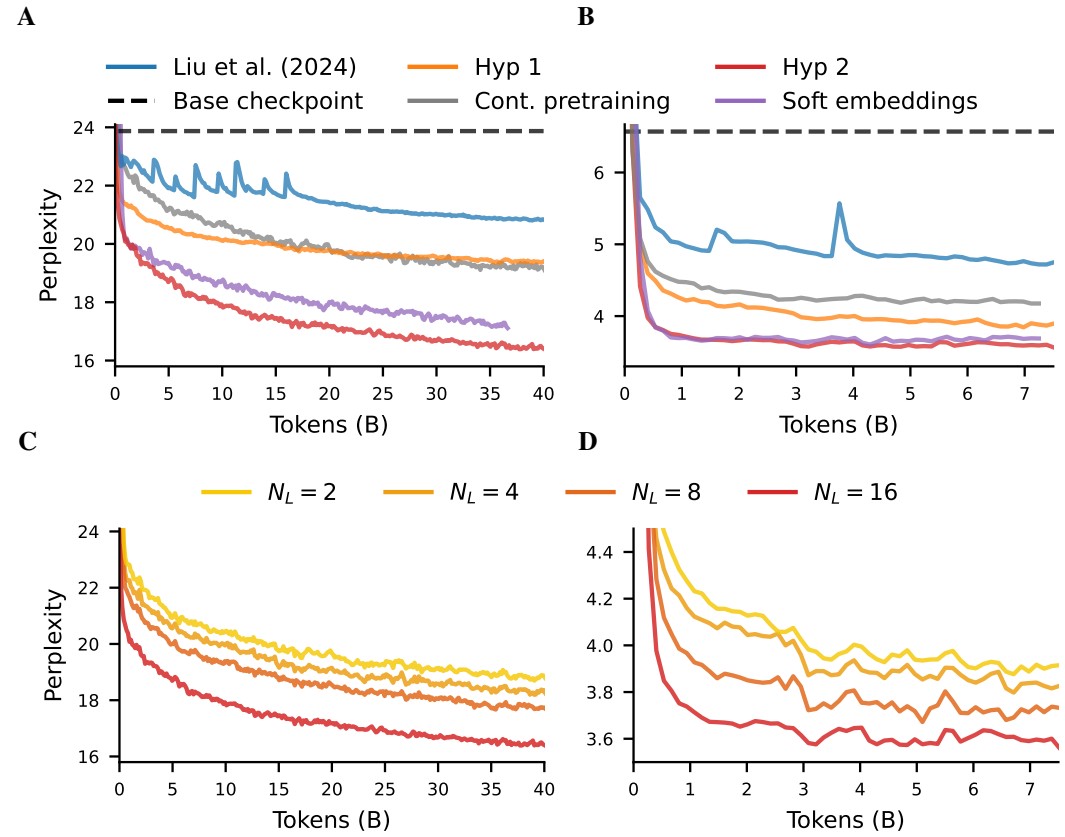

Figure 2: Validation perplexity whilst training on the FinWeb-Edu-100BT corpus. **A:** GPT-2 variants (using $N_L$=16 where applicable). **B:** Qwen-3 variants (using $N_L$=16 where applicable). **C:** Ablating number of latents $N_L$ of the GPT-2 Coprocessor for Hypothesis 2. **D:** Ablating number of latents $N_L$ of the Qwen-3 Coprocessor for Hypothesis 2.

**Baseline 2 - Base model + $N_L$ soft embeddings:** To test whether a single network can absorb the "System-2" role, we continue pretraining the base model alone while attaching the same number of new $N_L$ learnable soft tokens used by the Coprocessor setup (similar in spirit to Goyal et al., but with untied, slot-specific embeddings rather than a single shared token). This contrasts a unified versus dual-network design under identical data and latent embedding budgets. See App. B for visualization.

Note. Neither baseline is strictly parameter-matched—our dual-model has roughly twice as many weights—yet they provide useful sanity checks on data scaling and on the value of a separate Coprocessor.

### 4.1.2 RESULTS

**Qualitative replication of Liu et al. (2024b).** Substituting the 2-B parameter Gemma with much smaller GPT-2 (124 M) and Qwen-3 (0.6 B) still produces the qualitative trends reported by **?**(Fig. 2A/B). Perplexity (ppl) drops by 2 for GPT-2 and 2.5 for Qwen-3—an order-of-magnitude larger reduction than that reported in the original paper setup. On GPT-2 the dual-model also raises mean benchmark accuracy by +2.0 percentage point (pp) (Table 1). For Qwen-3, however, lower perplexity does not translate into higher benchmark scores; continued pretraining alone even degrades accuracy. Given Qwen-3's heavy post-training, such brittleness under distribution shift is expected. We therefore report all numbers relative to the continued-pretraining baseline.

**Both hypotheses outperform Liu et al. (2024b).** Relative to Liu et al.'s dual-model, *Hypothesis 1* (frozen Base, cache concatenation) improves average accuracy by +1.5 pp on GPT-2 and +4.4 pp on Qwen-3. *Hypothesis 2* (co-finetuned) improves by +3.7 pp (GPT-2) and +6.6 pp (Qwen-3). In ppl, the variants are ≈ 2 and ≈ 4 lower, respectively (Fig. 2A/B).

| Model | Hellaswag | ARC-Easy | Social IQA | PIQA | Winogrande |
|---|---|---|---|---|---|
| **GPT-2 variants** | | | | | |
| GPT-2 continued pretraining | 30.7 (+0.0) | 54.2 (+0.0) | 37.8 (+0.0) | 64.5 (+0.0) | 50.5 (+0.0) |
| GPT-2 + soft embeddings | 30.7 (+0.0) | 55.6 (+1.4) | 37.7 (−0.1) | 64.6 (+0.1) | 51.9 (+1.4) |
| Liu et al. (2024b) | 29.0 (−1.7) | 45.0 (−9.2) | 37.4 (−0.4) | 62.1 (−2.4) | 50.7 (+0.2) |
| Hypothesis 1 | 29.4 (−1.3) | 48.2 (−6.0) | **38.7 (+0.9)** | 62.9 (−1.6) | **52.3 (+1.8)** |
| Hypothesis 2 | **31.2 (+0.5)** | **55.8 (+1.6)** | 38.2 (+0.4) | **65.3 (+0.8)** | 52.1 (+1.6) |
| **Qwen variants** | | | | | |
| Qwen continued pretraining | 26.6 (+0.0) | 34.5 (+0.0) | 35.3 (+0.0) | 55.1 (+0.0) | 52.0 (+0.0) |
| Qwen + soft embeddings | 25.9 (−0.7) | 37.2 (+2.7) | 34.6 (−0.7) | 55.3 (+0.2) | 50.4 (−1.6) |
| Liu et al. (2024b) | 31.8 (+5.2) | 36.5 (+2.0) | 35.4 (+0.1) | 58.1 (+3.0) | 52.0 (+0.0) |
| Hypothesis 1 | 35.2 (+8.6) | 45.0 (+10.5) | 39.0 (+3.7) | 61.0 (+5.9) | **55.7 (+3.7)** |
| Hypothesis 2 | **37.3 (+10.7)** | **53.1 (+18.6)** | **41.9 (+6.6)** | **63.4 (+8.3)** | 51.1 (−0.9) |

Table 1: Model performance on standard small LLM benchmarks and $\Delta$ with continued pretraining baselines.

**Liu model versus data-matched continued pretraining.** Relative to our *data-matched* Baseline 1 (same number of training tokens), the Liu et al.'s dual-model exhibits higher ppl ($\approx +1.5$) and mixed accuracy: $-2.7$ pp on GPT-2 but $+2.1$ pp on Qwen-3. Averaged across families the net change is $\approx -0.3$ pp, underscoring the value of stricter baselines.

**Dual models versus the "soft-embedding" baseline.** Our second baseline—*Base model + $N_L$ soft embeddings*—keeps the architecture unified yet endows the network with the same latent budget. Against this control the dual-model results are underwhelming:

- Perplexity: *Hyp. 1* has higher ppl than the soft-embedding model for both families; *Hyp. 2* lowers it only marginally.
- GPT-2 benchmarks: Averaged over the five tasks in Table 1, *Hyp. 1* trails by $-1.8$ pp (46.3 vs. 48.1%), while *Hyp. 2* is only $+0.4$ pp higher (48.5%).
- Qwen-3 benchmarks: Dual models fare better: *Hyp. 1* $+6.5$ pp (47.2 vs. 40.7%); *Hyp. 2* $+8.7$ pp (49.4%).

**Summary and caveat.** At first glance both dual-model variants look successful: they outperform the **?** baseline and, on Qwen-3, yield sizeable benchmark gains. But the picture shifts once we introduce the *soft-embedding* control. That unified model matches the dual systems in token budget and latent capacity while using only half as many trainable weights, yet it equals or exceeds *Hyp. 1* and comes within a hair of *Hyp. 2*. This pattern indicates that the Coprocessor is not just "adding compute"—it is adding it *inefficiently*. A single LLM with the same aggregate parameter count indeed performs better, as we explicitly demonstrate in Appendix F. Accordingly, the next section turns to reasoning-specific benchmarks to see whether the dual architecture offers any benefit that a parameter-matched, soft-prompted model cannot already provide.

### 4.2 REASONING EVALUATION: BENCHMARKS AND A CONTROLLED STRESS TEST

In this section we wanted to test if additional latent tokens improve *reasoning* or do they mostly add compute? We first compare our systems on GSM8K Cobbe et al. (2021) and ProsQA Hao et al. (2024), then use a controlled *Countdown* stress test Pan et al. (2025) to probe robustness as combinatorial difficulty increases. We evaluate four systems: our reproduction of Liu et al., Hyp. 1 (frozen Base, cache-concat), Hyp. 2 (co-finetuned), and a single-model soft-embedding baseline with the same total latent budget $N_L$. For reference, we also compare against Coconut (GPT-2) Hao et al. (2024). More implementation details can be found in App. C.

**Benchmarks.** On GSM8K/ProsQA we follow the staged curriculum of Hao et al. (2024) and report accuracy after fine-tuning. Table 2 shows the best results ($N_L = 12$ for this task). Figure 3A plots *GSM8K* accuracy versus total latents $N_L$; the corresponding *ProsQA* curves are deferred to App. C, as this benchmark is near-saturated for competitive systems and adds limited insight.

**Stress test: Countdown.**   Countdown lets us scale difficulty in a controlled, task-homogeneous way by increasing the operand count (branching factor grows rapidly with each operand). Unlike the benchmarks, we train without curriculum and sweep both $N_L \in \{1, 2, 4, 8\}$ and operands $\in \{3, 4, 5\}$. An illustrative instance appears in the box below. Figure 3B plots accuracy vs. operands, combining GPT-2 and Qwen curves for clarity.

---

**Countdown example with operands = 4**

**User:** Using the numbers [19, 36, 55, 7], create an equation that equals 65.
**Assistant:** Let me solve this step by step.
```
<latent thinking>
<answer> 55 + 36 − 7 − 19 </answer>
```

---

**Findings.**   (i) *Rank order on GSM8K/ProsQA:* Hyp. 2 $\geq$ soft-embedding $\gg$ Hyp. 1 $\approx$ Liu (Table 2). (ii) *Scaling latents:* Fig. 3A shows accuracy is largely *flat* as $N_L$ increases and can dip at larger $N_L$. This contrasts with Section 4.1, where perplexity decreased as $N_L$ grew, suggesting that extra latents help next-token prediction but do not translate into more reliable reasoning. Interestingly, Coconut (GPT-2) exhibits the same dipping trend, indicating the difficulty is not specific to dual-model designs. (iii) *Stress test:* In Fig. 3B, increasing operands sharply reduces accuracy for all systems; larger $N_L$ helps slightly up to $N_L = 8$ but yields diminishing returns thereafter, and remains very close to the single-model soft-embedding baseline.

Table 2: Accuracy (%) on GSM8K and ProsQA after curriculum fine-tuning with $N_L = 12$.

| Model | GPT-2 | | Qwen-3 | |
| --- | --- | --- | --- | --- |
| | GSM8K | ProsQA | GSM8K | ProsQA |
| Coconut (B0) | **34.1** | 97.0 | – | – |
| Soft emb. (B1) | 26.5 | 97.7 | 38.5 | **99.5** |
| Liu et al. (B2) | 16.5 | 52.6 | 22.4 | 81.0 |
| Hyp. 1 (frozen, concat) | 12.0 | 54.1 | 24.0 | 79.0 |
| Hyp. 2 (co-finetuned) | 31.5 | **99.0** | **38.6** | **99.5** |

**Efficiency note (Hyp. 2 vs. Coconut).**   Coconut generates continuous thoughts sequentially with the same network; for a given input this requires approximately $N_L+1$ full forward passes (each time appending a new latent and re-processing), so compute and latency scale linearly with $N_L$ Hao et al. (2024). Our Hyp. 2 uses a strict three-pass schedule independent of $N_L$ (Base cache $\to$ Coprocessor latents $\to$ Base decode). Assuming similar model sizes, this yields an approximate forward-pass/FLOPs reduction of

$$\text{speedup} \approx \frac{N_L + 1}{3},$$

e.g., $\sim 5.7\times$ fewer full passes at $N_L{=}16$, while remaining batch-parallel (multiple augmentations per sequence can still be handled in one Coprocessor pass). In practice this removes Coconut's serial bottleneck—while achieving comparable task accuracy on GPT-2 (GSM8K: 31.5 vs. 34.1; ProsQA: 99.0 vs. 97.0; Table 2, Fig. 3a)—enabling larger models and datasets. However, it is worth noting that Hyp. 2 also introduces roughly $2\times$ as many trainable parameters as Coconut.

**Summary.**   Across reasoning benchmarks and the controlled stress test, adding latents mostly buys FLOPs, not robust reasoning. A unified soft-embedding model closely matches the best dual-model at equal $N_L$, and further increasing $N_L$ rarely helps—sometimes hurts. This motivates the interpretability analysis in §4.3.

### 4.3   INTERPRETABILITY: DO LATENTS SPECIALIZE OR COLLAPSE?

Reasoning can be viewed as composing *distinct* intermediate computations or modules Andreas et al. (2016); Lake & Baroni (2017); Veličković & Blundell (2021). If our Hypothesis 2 Coprocessor truly

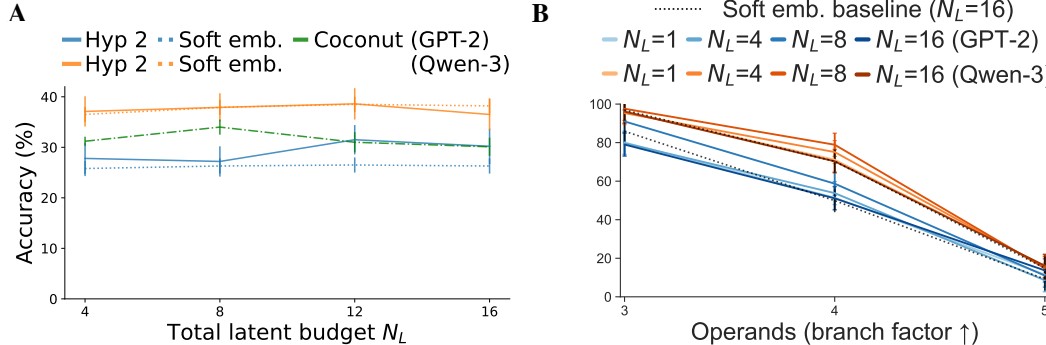

Figure 3: Ablating the latent budget. **A:** GSM8K accuracy vs. total latents $N_L$ (GPT-2 and Qwen; Coconut shown for GPT-2). **B:** Countdown accuracy vs. operands (3, 4, 5) with lines for $N_L \in \{1, 4, 8, 16\}$, merged across model families.

supports such division of labor, different *latents* should occupy meaningfully different directions in representation space. If, instead, latents mostly scale confidence without new algorithmic structure, they will reuse the same span, yielding *redundant* computations Olah et al. (2020); Elhage et al. (2022); Kornblith et al. (2019). We therefore analyze the **Coprocessor's last hidden layer** (the signal fed back as input embeddings to the Base) and test whether latents specialize.

**Diagnostic 1: Cross-capture heatmap.** *Intuition.* If latents specialize, the variance of latent $j$'s activations should not lie in latent $i$'s subspace. Let $X_i \in \mathbb{R}^{N_{eval} \times d}$ be the row-centered activations for latent $i$ collected across the full evaluation set (where $N_{eval}$ is the number of task-evaluation examples). For each $i$, we fit a minimal PCA subspace explaining at least $\tau = 97\%$ of its variance and form the projector $P_i$. We then quantify how much of latent $j$'s variance falls into that subspace,

$$H_{i,j} \;=\; \frac{\left\| X_j P_i \right\|_F^2}{\left\| X_j \right\|_F^2} \;\in [0,1],$$

visualize $H$ as a heatmap (note $H_{i,i} \geq \tau$), and summarize with the mean off-diagonal capture

$$\overline{H}_{\text{off}} \;=\; \frac{1}{N_L(N_L - 1)} \sum_{i \neq j} H_{i,j} \quad \text{(higher} \Rightarrow \text{more redundancy; lower is better).}$$

**Diagnostic 2: silhouette (cluster separation by latent).** We compute the silhouette score Kaufman & Rousseeuw (1990), $s \in [-1, 1]$, which for each point compares how close it is to its *own* latent cluster versus the *nearest other* latent cluster; higher $s$ means tighter, better separated clusters. This complements diagnostic 1: Instead of looking at directional variance reuse across latents (orientation overlap), it looks at instance-level spatial separation/cohesion of latent-labeled clusters. We report the global average (formal definition and the mean per-latent scores in App. §D).

### 4.3.1 RESULTS

**Large-scale training.** Figure 4a is nearly uniform and bright off the diagonal, and the quantitative scores confirm this collapse-like behavior: mean off-diagonal capture $\overline{H}_{\text{off}} = 0.9873$ (higher $\Rightarrow$ more redundancy), global silhouette $s = -0.1694$ with per-latent silhouettes mostly negative $(\{-0.26, \ldots, -0.07\}$; full vector in Appendix E). Together these indicate that occurrences labeled by different latents largely occupy the *same* subspaces and are not cluster-separated.

**GSM8K with curriculum.** Relative to the large-scale pretraining task, fine-tuning induces a noticeable reduction in directional redundancy ($\overline{H}_{\text{off}}$ drops from 0.987 to 0.914). While the global silhouette change is slim, it marks a qualitative shift, crossing from negative ($-0.05$ without curriculum) to positive (0.019 with curriculum). This geometric reorganization coincides with a substantial performance gain for Hypothesis 2, improving from 11.2% (without curriculum) to 31.5%. This suggests that the curriculum induces an important shift for reasoning even if the specific dynamics of this shift remain to be fully characterized.

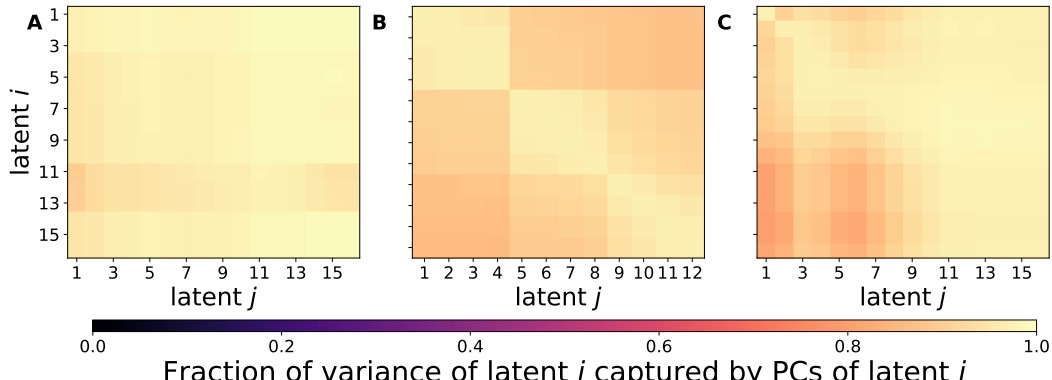

Figure 4: **Latent cross-subspace capture heatmaps** (last Coprocessor layer). Each cell $(i, j)$ shows the fraction of variance of latent $j$ captured by the principal subspace of latent $i$. **A:** Large-scale training. **B:** Fine-tuning on GSM8k with curriculum. **C:** Finetuning on countdown with operands = 4.

**Countdown (operands = 4).** Countdown induces substantially better cluster separation (global silhouette $s = 0.4531$), yet the cross-subspace redundancy remains high with $\overline{H}_{\text{off}} = 0.9382$. This realizes the "*separated but redundant spans*" regime: clusters are distinct in Euclidean space (high silhouette) but their principal subspaces still explain most of each other's variance (high off-diagonal capture). Interestingly, Fig 4c shows distinctiveness degrading beyond latent 8, aligning with Fig 3B where accuracy improves from $N_L \in \{1, 4, 8\}$ but drops at $N_L = 16$.

Across all three settings the latents tend toward redundant computations: their subspaces are highly overlapping (high $\overline{H}_{\text{off}}$), even when the data task encourages class separation (Countdown). Task-aligned fine-tuning helps (curriculum < pretraining in $\overline{H}_{\text{off}}$), but not enough to yield "separated and specialized" latents (high silhouette and low off-diagonals).

### 4.4 CAN EXPLICIT REGULARIZATION FIX LATENT COLLAPSE?

To address the representational redundancy observed in §4.3 and test if lack of diversity hinders reasoning, we introduce an auxiliary orthogonality loss. We add this penalty to the standard next-token prediction objective: $\mathcal{L}_{total} = \mathcal{L}_{NTP} + \lambda \mathcal{L}_{orth}$.

Let $Z \in \mathbb{R}^{N_L \times d}$ be the L2-normalized latent representations. We penalize the mean squared cosine similarity between distinct latents to force them into orthogonal directions:

$$\mathcal{L}_{orth} = \frac{1}{N_L(N_L - 1)} \sum_{i \neq j} (Z_i Z_j^T)^2 \tag{1}$$

**Recovering Systematic Scaling in Reasoning (Countdown).** In the combinatorial Countdown task (operands=4), explicit regularization yields a distinct phase shift. As shown in Figure 5C, a high penalty ($\lambda = 3$) forces the latents into specialized, non-overlapping subspaces (Silhouette: 0.99, $\overline{H}_{\text{off}}$: 0.55). Crucially, this geometric separation translates to downstream performance: accuracy improves by 11.5 pp at $N_L = 16$ (62.6% vs 51.1%). Most importantly, Figure 5B shows that regularization restores *monotonic scaling* with respect to the latent budget $N_L$. Unlike the unregularized baseline where adding latents eventually hurt performance, the regularized model effectively utilizes the additional capacity, validating that the Coprocessor *can* perform structured reasoning if collapse is prevented. The reader can find the evaluation accuracy score throughout training in App H (Figure 16).

**The Trade-off with General Modeling.** This specialization, however, appears antagonistic to large-scale language modeling. In large-scale pretraining (Figure 5A), increasing $\lambda$ degrades validation perplexity. This suggests that for general language distributions, the model prefers using latents as redundant "bandwidth" to propagate confident predictions (a semantic shortcut) rather than distinct reasoning steps. Consequently, the "System 2" structure required for reasoning might be locally

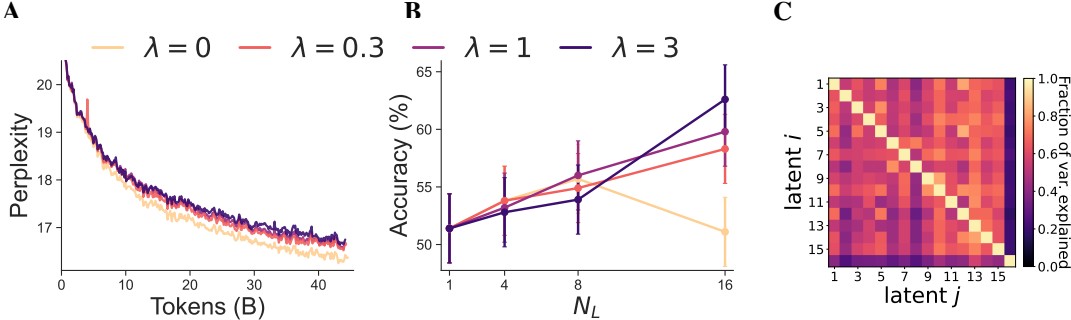

Figure 5: Explicit Latent Regularization Results. **A:** Large scale training validation perplexity (lower is better) worsens as regularization $\lambda$ increases. **B:** Countdown (operands=4) accuracy restores monotonic scaling with $N_L$ under strong regularization ($\lambda = 3$). **C:** Latent cross-capture heatmap for Countdown ($\lambda = 3$) shows distinct, non-overlapping subspaces (compare to Fig 4).

suboptimal for pretraining, creating a tension where the objective naturally favors the "System 1" geometry of redundancy.

**Interaction with Curriculum (GSM8K).** On GSM8K, we found no systematic improvement ($30.4 \pm 1.4\%$ across $\lambda \in [0, 3]$). We hypothesize this is because the curriculum already imposes a "block-wise" structure (grouping latents by reasoning steps), which may conflict with the atomized, token-wise orthogonality enforced by $\mathcal{L}_{orth}$. Future work must explore objectives that encourage diversity *between* reasoning steps while allowing redundancy *within* a computational block.

## 5 DISCUSSION

Building on Liu et al. (2024b), we cast the coprocessor architecture as a principled attempt to disentangle "System-1" token prediction from "System-2" abstract reasoning. Treating latent embeddings as a private communication channel between the two modules clarifies the conceptual link to dual-process theories in cognitive science and motivates our two new training hypotheses. However, in our setting, simply providing the channel (as in Liu et al. (2024b)) or strengthening it (our H1/H2) did not induce System-2-like computation.

Replacing last-layer hidden-state injection with full KV-cache concatenation yields modest perplexity and benchmark gains over the original design, yet falls short once stricter baselines are introduced. Its benefits appear sensitive to model family and vanish on harder reasoning tasks. Jointly updating both models produces the strongest results across all experiments, confirming that bidirectional adaptation facilitates cross-model communication. However, a *unified* network trained with the same soft-token budget narrows—sometimes erases—the gap, suggesting that current dual-system instantiations add compute inefficiently rather than unlocking qualitatively new reasoning abilities.

The soft-embedding baseline, parameter-matched continued pre-training, and Coconut all highlight scenarios where dual-model gains either disappear or can be matched closely by simpler means. Our Countdown experiments further show that scaling the latent budget beyond eight tokens fails to deliver systematic robustness as combinatorial complexity explodes. Our interpretability analysis seems to corroborate this by showing that learned latents largely occupy overlapping representational subspaces: extra latents mostly amplify confidence rather than add new algorithmic structure.

Our results do not invalidate the dual-system framework; they indicate that *how* the two models exchange information remains an open problem, and that current instantiations do not create conditions for System-2-like computation to emerge. Our preliminary experiments on latent regularization and curriculum learning support this view, suggesting promising directions by (i) designing objectives that explicitly reward diversity or orthogonality in latent representations to encourage broader search, and (ii) developing training schedules that preserve large-scale language competence while gradually shaping latent spaces for multi-step reasoning.

## REPRODUCIBILITY STATEMENT

Section 4.1 specifies the datasets, compute/token budgets, base models, and hyperparameters used, plus our evaluation protocol. Appendix Fig. 7 details the three-pass implementation that supports multiple augmentations per sequence while separating latent computation from next-token prediction. As some implementation details in Liu et al. (2024b) were not fully specified and a bit unclear, we document our implementation to make these dual-model architectures reproducible and to foster further analysis and understanding from the community.

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

# APPENDIX

## A    USE OF LARGE LANGUAGE MODELS

The language in this paper was at times polished with the assistance of an LLM. The model was not used for research ideation, experimental design, or data analysis.

## B    LARGE SCALE TRAINING DETAILS

### B.1    DETAILED FORMULATION OF THE TRAINING OBJECTIVE

Here we formalize the three-pass process described in Section 3. Let $x = (x_1, \ldots, x_S)$ be a sequence. We select a set $\mathcal{T} = \{t_1, \ldots, t_M\}$ of $M$ augmentation indices, sampled uniformly from valid sequence positions.

**(i) KV-cache generation:** We compute the frozen Key-Value cache for the entire sequence once: $KV_\theta(x) = B_\theta(x)$.

**(ii) Latent augmentation:** For each augmentation site $t_m \in \mathcal{T}$, the Coprocessor generates a sequence of latents $Z^{(m)}$. It attends to the Base cache prefix $KV_\theta(x_{<t_m})$ and $N_L$ learnable soft tokens $S_{soft}$:

$$Z^{(m)} = C_\phi(KV_\theta(x_{<t_m}) \oplus S_{soft}) \in \mathbb{R}^{N_L \times d} \tag{2}$$

**(iii) Decoding:** The Base model predicts the next $N_A$ tokens (the "ahead" tokens) conditioned on the injected latents. The function inject$(\cdot)$ abstracts the specific interaction mechanism (embedding injection for Hyp. 2 or cache concatenation for Hyp. 1). The loss is computed only over these $N_A$ tokens for every augmentation site $m$, maximizing the likelihood:

$$\mathcal{L} = \sum_{m=1}^{M} \sum_{j=1}^{N_A} \log p_\theta \left( x_{t_m+j} \mid \text{inject}\left(KV_\theta(x_{<t_m}), Z^{(m)}\right), x_{t_m:t_m+j-1} \right) \tag{3}$$

$M$ allows us to train on multiple randomly sampled reasoning segments per sequence in parallel, while $N_A$ defines the window of future tokens that provide the supervisory signal.

### B.2    IMPLEMENTATION DETAILS

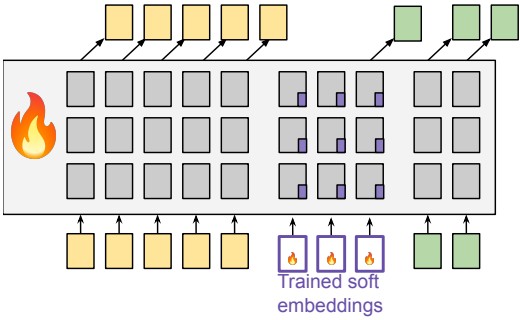

Figure 6: **Soft-embedding baseline (main control).** A single transformer (no Coprocessor) plays both "System-1" and "System-2" roles. We attach $N_L$ learnable soft tokens ("latent slots") to the Base and continue pre-training the same Base checkpoint LLM and these tokens on the same dataset and token budget as the dual-model runs. This matches the dual setup's latent capacity ($N_L$) while using half the trainable parameters, directly testing whether a separate coprocessor—and its cache-level latent communication with the Base—provides benefits beyond added compute. During decoding, the learned soft tokens serve as the latent context via input-embedding injection.

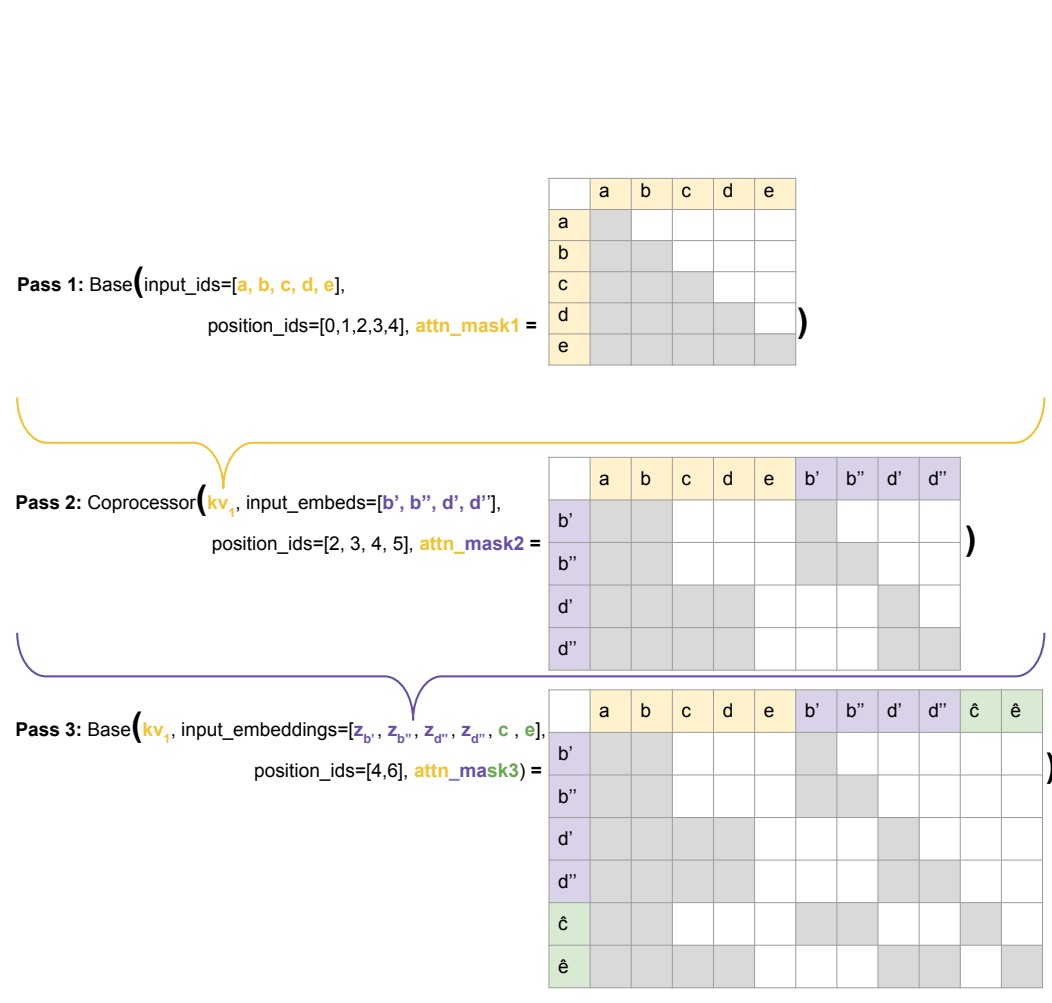

Figure 7: **Three-pass schedule on a toy sequence.** Example with input "`abcde`" and two augmentation sites at b and d ($M=2$), $N_L=2$ latent slots per site (b', b'', d', d''), and $N_A=1$ ahead token per site. *Pass 1* (**Base**): run once on the raw sequence to form the cache $KV_\theta(x)$. *Pass 2* (**Coprocessor**): consume $KV_\theta(x)$ together with the *concatenated* $M \times N_L$ latent placeholders [b', b'', d', d''], producing per-slot latent embeddings $z_{b'}, z_{b''}, z_{d'}, z_{d''}$. *Pass 3* (**Base** decode): reuse the same $KV_\theta(x)$ and feed the *concatenated* outputs plus all $M \times N_A$ ahead tokens (one per site in this toy) as input embeddings (e.g., $[z_{b'}, z_{b''}, z_{d'}, z_{d''}, \hat{c}, \hat{e}]$). Concatenation in Passes 2–3 enables many augmentations per sequence to be trained in parallel while keeping latent computation (Pass 2) separated from next-token prediction (Pass 3). We are still unsure by the single-pass implementation sketched by Liu et al. (2024b) (no public code), but this schedule preserves disentanglement without sacrificing batch parallelism.

## C    REASONING IMPLEMENTATION DETAILS

**Benchmarks and curriculum.**    For GSM8K Cobbe et al. (2021) and ProsQA Hao et al. (2024) we follow the staged curriculum of Hao et al. (2024). At stage $k$, the first $k$ CoT steps are replaced by $k \times c$ continuous latents. Loss is masked on question tokens and on the latent spans; only the remaining language tokens contribute to the objective. We sweep $c \in \{1, 2, 3, 4\}$ with $k=4$, so the total latent budget is $N_L = k\,c \in \{4, 8, 12, 16\}$. All benchmark fine-tuning runs use a learning rate of $1\times10^{-4}$. To make the comparison to Hao et al. (2024) as faithful as possible, we deliberately minimize changes to their curriculum recipe; better schedules may exist for our dual-architecture, but exploring them is out of scope here.

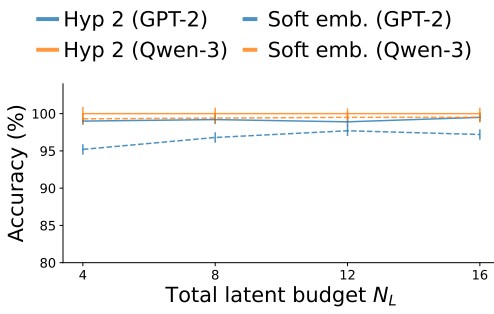

*Notes.* ProsQA is near-saturated for both GPT-2 and Qwen-3, so curves are flat across $N_L$ and the soft-embedding baseline tracks Hypothesis 2 closely. This complements Fig. 3A (GSM8K), where accuracy likewise fails to grow monotonically with $N_L$. Plotting conventions match the main text; dotted lines denote the single-model soft-embedding baseline.

Figure 8: ProsQA accuracy vs. total latents $N_L$.

**Countdown setup.**    Countdown is trained *without* curriculum. Operands are sampled uniformly from $[1, 50]$ and the target from $[0, 100]$; expressions use $+, -, \div$ and $\times$ (as in TinyZero Pan et al. (2025)). For each (operands, $N_L$) setting we train for 3 epochs over 262,144 generated training samples and evaluate on 1,024 held-out samples. We vary the operand count $\in \{3, 4, 5\}$ and latent budget $N_L \in \{1, 2, 4, 8\}$. All runs use a learning rate of $1\times10^{-4}$. The branching factor grows with operands as $\mathrm{Catalan}(n-1)\, 2^{n-1}\, n!$, yielding the controlled difficulty axis reported in Fig. 3B.

**Fine-tuning from large-scale pre-training.**    On GSM8K/ProsQA, resuming from large-scale pre-training *hurts* relative to starting from base checkpoints (Tables 3–4), suggesting a mismatch between next-token pre-training geometry and our curriculum-based supervision at this scale.

**Comparison protocols (all end curriculum at $N_L=16$).**

- *Hyp. 2 (scratch):* co-finetune Base and Coprocessor from the standard base checkpoints; staged curriculum with $k=4$ stages and $c=4$ latents per stage.
- *Resume + curriculum:* initialize Hyp. 2 from the large-scale training checkpoints (Sec. 4.1) and run the same curriculum ($k=4$, $c=4$).
- *Straight-to-16:* initialize from the large-scale checkpoints but skip the curriculum; a single stage with $k=1$, $c=16$.

Table 3: Effect of resuming from large-scale pre-training on **GSM8K**. Accuracy (%); $\Delta$ is the change vs. Hyp. 2 (scratch).

| Model | Hyp. 2 (scratch) | Resume + curriculum | Straight-to-16 |
|---|---|---|---|
| GPT-2 | 31.5 | 26.0 ($\Delta - 5.5$) | 28.2 ($\Delta - 3.3$) |
| Qwen-3 | 38.6 | 35.7 ($\Delta - 2.9$) | 37.3 ($\Delta - 1.3$) |

By contrast, on Countdown (no curriculum) resuming helps at $N_L=16$: Fig. 9 compares three curves per family (GPT-2 left, Qwen-3 right)—Hypothesis 2 (from scratch), Hypothesis 1, and Hypothesis 2 resumed from large-scale training—and the resumed Hypothesis 2 dominates. One plausible explanation is that Countdown's objective is closer to the pre-training signal (no curriculum,

Table 4: Effect of resuming from large-scale pre-training on **ProsQA**. Accuracy (%); $\Delta$ is the change vs. Hyp. 2 (scratch).

| Model | Hyp. 2 (scratch) | Resume + curriculum | Straight-to-16 |
|---|---|---|---|
| GPT-2 | 99.0 | 97.1 ($\Delta - 1.9$) | 97.3 ($\Delta - 1.7$) |
| Qwen-3 | 99.5 | 88.3 ($\Delta - 11.2$) | 84.5 ($\Delta - 15.0$) |

homogeneous task family), whereas GSM8K/ProsQA curriculum alters the supervision structure in a way that conflicts with the pre-trained latent geometry. A fuller exploration of how to transfer large-scale pre-training into reasoning-specific curricula is a promising direction for future work.

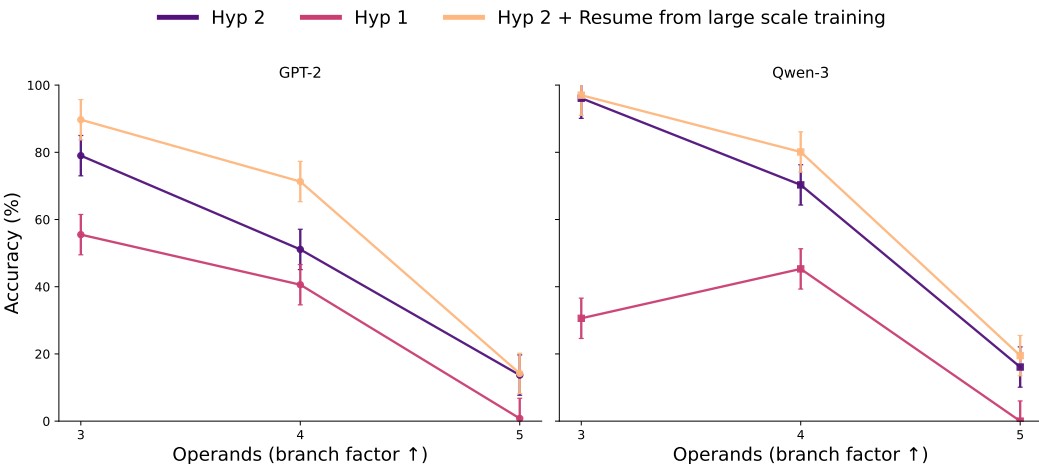

Figure 9: Countdown appendix: accuracy at $N_L=16$ for Hypothesis 2 (from scratch), Hypothesis 1, and Hypothesis 2 resumed from large-scale pre-training. Left: GPT-2. Right: Qwen-3. Resuming improves Countdown despite hurting GSM8K/ProsQA, hinting at a mismatch between curriculum-based supervision and pre-trained latent geometry.

## D  SILHOUETTE DEFINITION

Let $X = \{x_p\}_{p=1}^N \subset \mathbb{R}^d$ be all occurrences from the coprocessor's last hidden layer, labeled by latent indices $y_p \in \{1, \ldots, N_L\}$ with clusters $C_\ell = \{p : y_p = \ell\}$. For $p \in C_\ell$, define the mean intra-latent distance

$$a_p = \frac{1}{|C_\ell| - 1} \sum_{\substack{q \in C_\ell \\ q \neq p}} \|x_p - x_q\|_2,$$

and, for $\ell' \neq \ell$, the mean distance

$$d(p, C_{\ell'}) = \frac{1}{|C_{\ell'}|} \sum_{q \in C_{\ell'}} \|x_p - x_q\|_2, \qquad b_p = \min_{\ell' \neq \ell} d(p, C_{\ell'}).$$

The silhouette of $p$ is

$$s_p = \frac{b_p - a_p}{\max\{a_p, b_p\}} \in [-1, 1].$$

We report the global silhouette $s = \frac{1}{N} \sum_{p=1}^N s_p$ and the per-latent silhouette $s_\ell = \frac{1}{|C_\ell|} \sum_{p \in C_\ell} s_p$. For singleton clusters, we set $s_p = 0$.

# E  ADDITIONAL INTERPRETABILITY RESULTS

## E.1  EFFECTIVE SUBSPACE DIMENSIONS

To verify that the high cross-capture values ($\bar{H}_{off}$) observed in Figure 4 are not artifacts of high-dimensional subspaces trivially covering the ambient space (where $P_i \approx I$), we analyze the dimensionality of the PCA subspaces used in our metric.

For each latent $i$, we compute the rank $k$ of the projector $P_i$ required to explain $\tau = 97\%$ of the variance. Table 5 reports the average rank across all latents for the GPT-2 backbone ($d_{model} = 768$).

Table 5: Average dimensionality of latent PCA subspaces (explaining 97% variance) for GPT-2 ($d = 768$).

| Task | Avg. Subspace Rank | % of Ambient Dim |
|---|---|---|
| Large Scale Pretraining | $\approx 2$ | 0.2% |
| Countdown | $\approx 60$ | 7.8% |
| GSM8K (Curriculum) | $\approx 161$ | 20.9% |

In all settings, the effective subspace dimension is significantly smaller than the ambient dimension. This confirms that the high off-diagonal capture we observe implies specific, redundant directional alignment between latents, rather than broad spatial coverage.

## E.2  MAIN PAPER TASKS

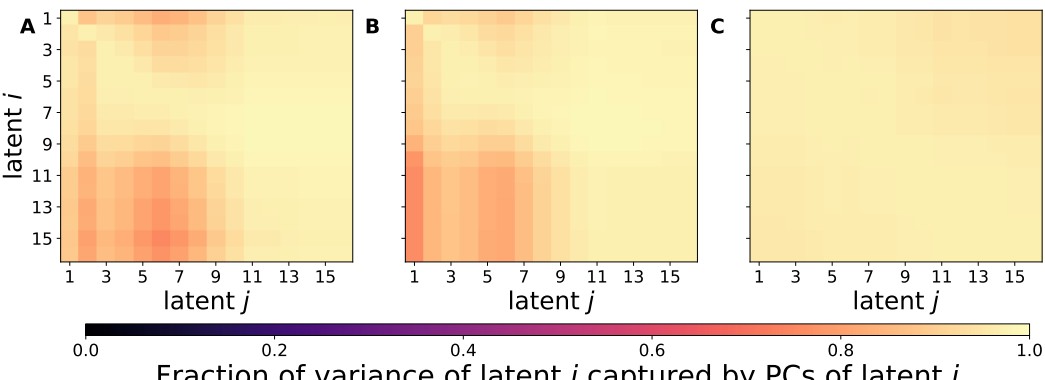

Figure 10: **Supplementary latent cross-subspace capture heatmaps** (coprocessor last layer). **A:** *Countdown* with operands = 3. **B:** *Countdown* with operands = 5. **C:** *GSM8K* without curriculum. Panels A/B are qualitatively similar to the main-paper operands = 4 plot; panel C shows stronger collapse than its curriculum counterpart.

| Latent | Large-scale pretrain | GSM8K (curr.) | Countdown (op=4) |
|---|---|---|---|
| 1 | -0.158 | -0.014 | 0.912 |
| 2 | -0.089 | -0.036 | 0.506 |
| 3 | -0.176 | -0.036 | 0.324 |
| 4 | -0.173 | -0.032 | 0.209 |
| 5 | -0.259 | -0.044 | 0.243 |
| 6 | -0.113 | -0.047 | 0.550 |
| 7 | -0.067 | -0.047 | 0.635 |
| 8 | -0.143 | -0.045 | 0.629 |
| 9 | -0.117 | -0.032 | 0.723 |
| 10 | -0.170 | -0.040 | 0.742 |
| 11 | -0.202 | -0.022 | 0.568 |
| 12 | -0.252 | 0.019 | 0.357 |
| 13 | -0.193 | – | 0.336 |
| 14 | -0.133 | – | 0.461 |
| 15 | -0.191 | – | 0.026 |
| 16 | -0.211 | – | 0.030 |

Table 6: **Per-latent silhouette** $s_\ell$. Dashes indicate unused latents in the GSM8K run (here $N_L$=12). Large-scale pretraining shows uniformly negative silhouettes (cluster intermixing); GSM8K with curriculum hovers near zero (weak separation); Countdown exhibits strong separation for most latents while still showing redundancy in cross-subspace capture (see main-text heatmaps).

### E.3 EXTRA REASONING TASKS: SVAMP AND MULTIARITH

**Setup.** We fine-tuned the Qwen-3 dual-model variant (Hypothesis 2) on the training sets of these tasks. Unlike the GSM8K experiments in the main text (which utilized a staged curriculum derived from CoT data), these models were fine-tuned directly on the target data without a curriculum. We report results exclusively for Qwen-3; we found that the smaller GPT-2 model failed to improve over random baselines on these datasets. We attribute this to the limited number of training examples available in SVAMP and MultiArith compared to Countdown, and the absence of a CoT-based curriculum which was essential for stabilizing optimization on GSM8K.

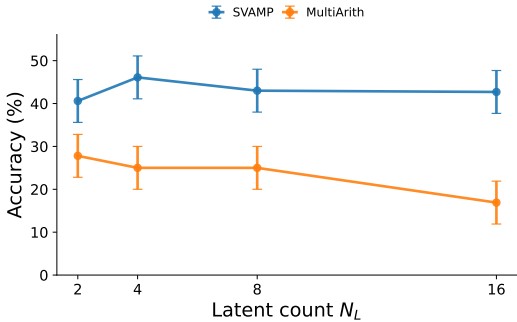

Figure 11: Accuracy scaling with respect to latent count $N_L$ on SVAMP and MultiArith (Qwen-3) after finetuning on 3 training epochs. Similar to GSM8K and Countdown, we observe no systematic benefit from increasing the latent budget; performance either plateaus (SVAMP) or degrades (Multi-Arith), consistent with the hypothesis that the latents are suffering from representational collapse.

**Results: Latent Collapse.** Despite learning the tasks to a moderate degree, the latent space analysis at $N_L = 16$ reveals the same pattern of collapse observed in the main paper:

- **MultiArith:** The latents exhibit high redundancy, with a mean off-diagonal capture of $\overline{H}_{off} = 0.950$ and a global silhouette score of $s = -0.0114$.

- **SVAMP:** Similarly, we observed a mean off-diagonal capture of $\overline{H}_{off} = 0.949$ and a global silhouette score of $s = -0.056$.

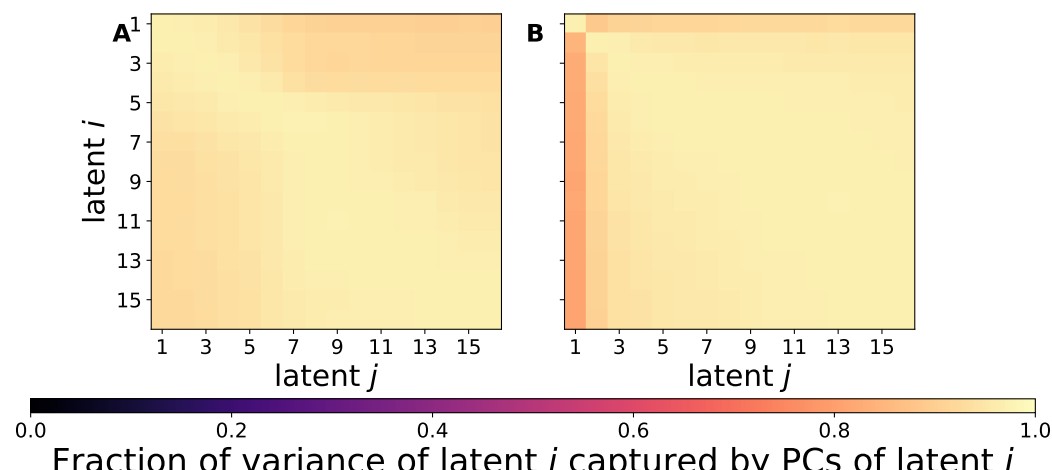

Figure 12: Latent cross-subspace capture heatmaps for **A)** SVAMP and **B)** MultiArith (Qwen-3). Both tasks display high off-diagonal brightness ($\overline{H}_{off} \approx 0.95$), indicating that the "latent collapse" phenomenon generalizes to other arithmetic reasoning benchmarks.

To address the question of whether the observed representational redundancy is specific to our main datasets or a general property of the architecture, we extended our interpretability analysis to two additional arithmetic reasoning benchmarks: SVAMP and MultiArith.

**Results: Latent Scaling.** We swept the latent budget $N_L \in \{2, 4, 8, 16\}$ to test if additional latent capacity translates to better reasoning. As shown in Figure 11, we observe the same "flat or degrading" scaling law seen in the main text:

- **SVAMP:** Performance remains largely stagnant as $N_L$ increases, peaking slightly at $N_L = 4$ ($\approx 45\%$) before flattening out around $42\%$.
- **MultiArith:** Accuracy actually degrades as latent capacity increases, dropping from $\approx 28\%$ at $N_L = 2$ to $< 20\%$ at $N_L = 16$.

This confirms that without explicit regularization or curriculum to shape the latent space, simply adding more "thinking tokens" does not yield robust performance gains.

## F  GPT-2 WITH MATCHED TRAINING PARAMETERS

To strictly verify whether the dual-model architecture offers benefits beyond simply increasing the parameter count, we compare our 124M dual-model setup against a single, larger GPT-2 model.

**Setup.** We train a "small to medium" GPT-2 with 16 layers, 16 heads, and 1024-dimensional embeddings, resulting in approximately 254M parameters. This matches the aggregate parameter count of our dual-system (124M Base + 124M Coprocessor). We first pre-train this model from scratch using the NanoGPT framework (Karpathy, 2022) to serve as a parameter-matched baseline. We note that we limited this ablation to the GPT-2 family, as performing full pre-training of a parameter-matched Qwen model ($\approx$ 1.2B parameters) from scratch is unrealistic for the scope of this paper.

**Results.** As shown in Figure 13, the parameter-matched single model (Soft embeddings 250M) attains lower validation perplexity than the Hypothesis 2 dual-system.

**A Conservative Comparison.** We acknowledge that this comparison places the dual-model system at an initialization disadvantage. While the Base component is pre-trained, the Coprocessor (comprising 50% of the aggregate parameters) is initialized randomly and must learn from scratch. In contrast, the single 250M model is trained as a fully coherent unit. This is our best attempt at having a matched parameter baseline.

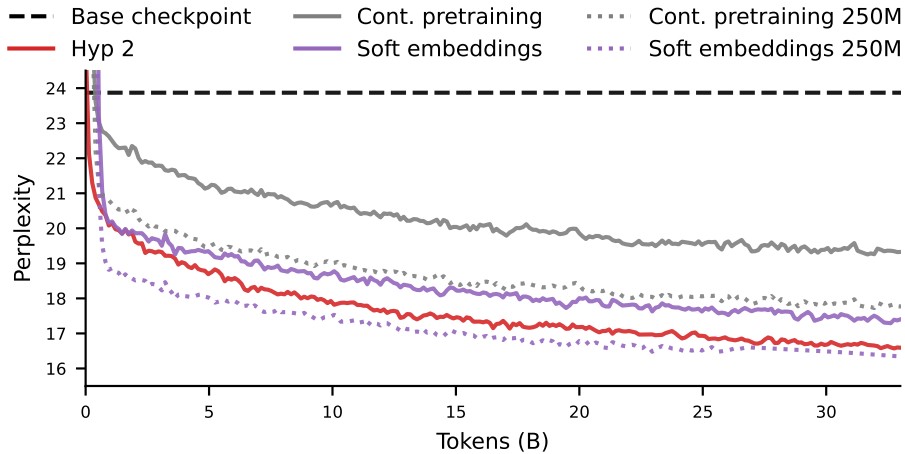

Figure 13: **Validation perplexity for GPT-2 variants vs. a parameter-matched single model.** We compare the 124M dual-model runs against a single 250M GPT-2 (dotted lines). The single model attains lower perplexity, suggesting that the dual architecture is an inefficient way to scale parameters compared to a unified model of the same size.

## G  EXTRA ABLATIONS

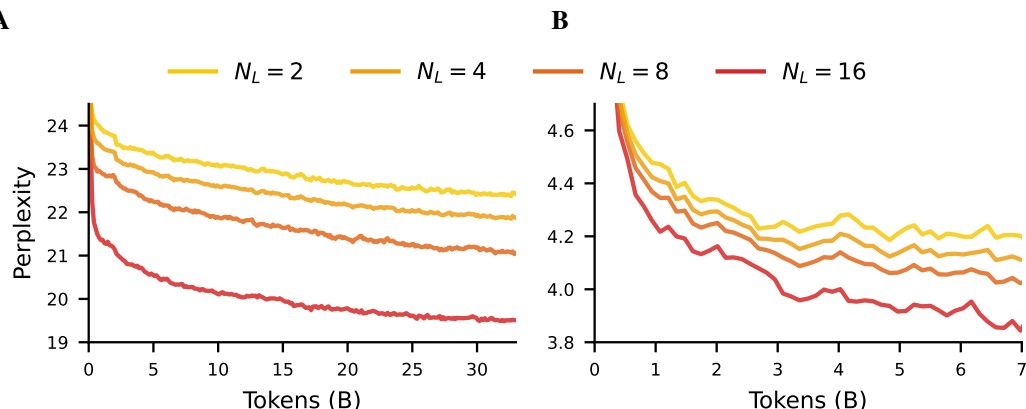

Figure 14: Validation perplexity whilst training on the FinWeb-Edu-100BT corpus. **A:** Ablating number of latents $N_L$ of the GPT-2 Coprocessor for Hypothesis 1. **B:** Ablating number of latents $N_L$ of the Qwen-3 Coprocessor for Hypothesis 1.

| Model | Cont. pretraining | $N_A = 32$ | $N_A = 16$ | $N_A = 8$ | $N_A = 4$ |
|---|---|---|---|---|---|
| GPT-2 | 30.90 | **31.29** (+0.39) | 31.25 (+0.35) | 31.04 (+0.14) | 30.93 (+0.03) |
| Qwen-3 | 34.50 | 52.20 (+17.70) | **53.10** (+18.60) | 48.20 (+13.70) | 46.80 (+12.30) |

Table 7: ARC-Easy accuracy (in %) for varying numbers of ahead tokens $N_A$ during coprocessor training. For GPT-2, $N_A = 32$ achieves the highest accuracy (31.29, +0.39 points over the *Cont. pretraining* baseline of 30.90). For Qwen-3, $N_A = 16$ achieves the highest accuracy (53.10, +18.60 points over the *Cont. pretraining* baseline of 34.50).

### G.1  LENGTH GENERALIZATION ANALYSIS

To address concerns regarding the robustness of our proposed architectures—specifically whether the models are sensitive to sequence length scaling—we evaluate our trained models on sequence lengths extending beyond the training context.

All models were trained with a sequence length of $S = 512$ despite the model supporting up to 1024. This design choice ensures that our generalization tests (up to $2 \times S = 1024$) isolate the effect of our training method from the absolute positional limits of the pre-trained checkpoint. We evaluate validation perplexity on the FineWeb-Edu corpus with sequence lengths scaled by factors of $\{1.0\times, 1.125\times, 1.25\times, 1.5\times, 2.0\times\}$ (evaluated up to 1024 tokens).

The results are summarized in Figure 15:

- **Hypothesis 1 (Frozen-Base KV Augmentation):** Contrary to concerns that cache concatenation might disrupt attention mechanisms at longer lengths, Hypothesis 1 demonstrates strong robustness. For Qwen-3, performance remains stable across all tested lengths. For GPT-2, the model generalizes well up to $1.5\times$ length, with only slight degradation observed at $2\times$ length.

- **Hypothesis 2 (Co-finetuned):** The fully finetuned dual-model system shows architecture-dependent generalization. On Qwen-3 (which uses Rotary Positional Embeddings), H2 generalizes effectively. However, on GPT-2 (which uses absolute positional embeddings), H2 fails to generalize, exhibiting rapid perplexity degradation when inputs exceed the training context.

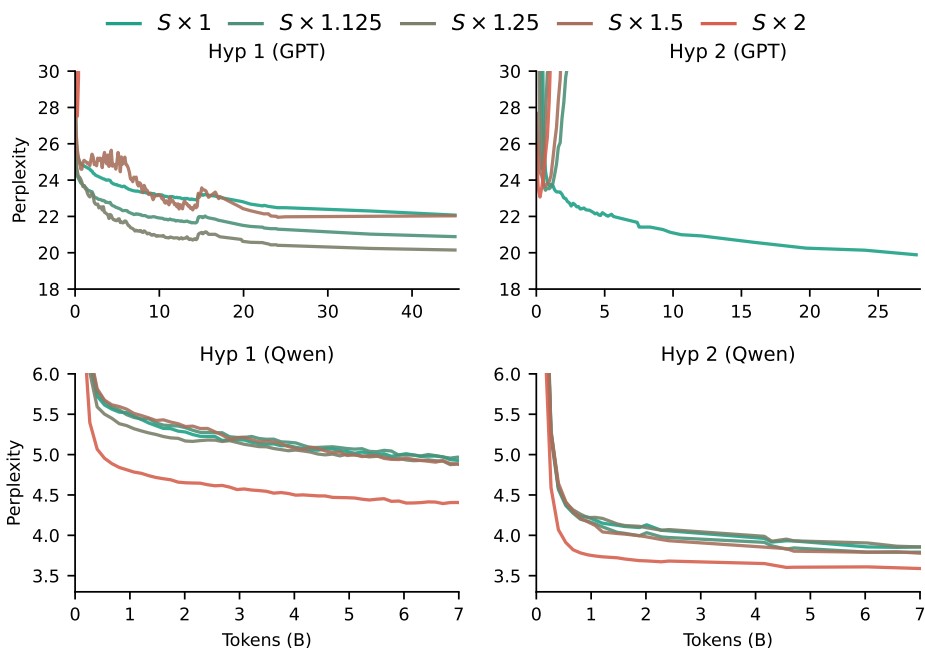

Figure 15: Length generalization tests evaluating validation perplexity on sequence lengths up to $2\times$ the training context ($S = 512$). **Top Row:** GPT-2 variants. **Bottom Row:** Qwen-3 variants. **Left Column:** Hypothesis 1 (Frozen Base + KV Concat). **Right Column:** Hypothesis 2 (Co-finetuning).

This suggests that these dual-model architectures are robust to length generalization provided they utilize modern relative positional embeddings (like RoPE in Qwen-3), whereas architectures with absolute positional embeddings (as in GPT-2) may struggle to generalize in the co-finetuned setting (Hyp 2).

# H   TRAINING DYNAMICS OF REGULARIZATION

To understand whether the orthogonality loss impacts convergence speed or stability, we track the evaluation accuracy on the Countdown task (operands=4) throughout the training process. Figure 16 illustrates the learning curves for varying regularization strengths $\lambda \in \{0, 0.03, 0.3, 1, 3\}$.

We observe that while strong regularization ($\lambda = 3$) initially penalizes performance, leading to a slower start compared to the unregularized baseline ($\lambda = 0$), it prevents the early saturation seen in the baseline. The unregularized model converges rapidly to a lower performance ceiling, likely settling into a redundant local minimum. In contrast, the regularized models continue to improve later into training.

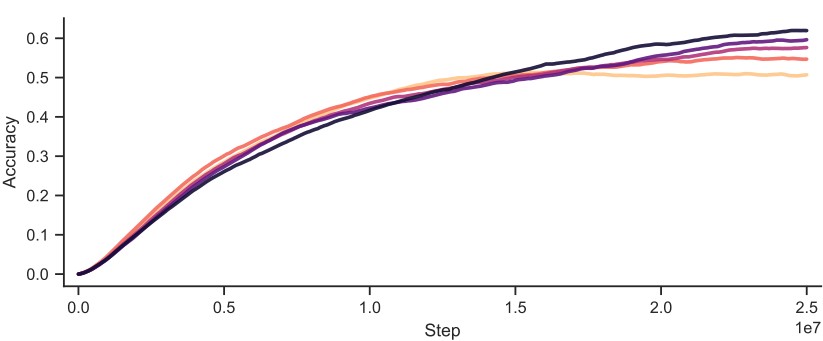

Figure 16: Countdown evaluation accuracy (Operands=4 for $N_L = 16$) tracked over training steps for different orthogonality regularization strengths $\lambda$. While the baseline ($\lambda = 0$) converges quickly, stronger regularization ($\lambda = 3$) sustains improvement for longer, ultimately reaching higher accuracy.

