# OpenReview forum: "Exploring System-1 and System-2 Communication for Latent Reasoning in LLMs"
_ICLR.cc/2026/Conference — Submitted to ICLR 2026_

### Official Review · Reviewer_Pkeg · 2025-10-17

**Soundness:** 3
**Presentation:** 3
**Contribution:** 3
**Rating:** 4
**Confidence:** 3

**Summary:**

This paper revisits dual-architecture latent reasoning for large language models (LLMs), where a “Base” model exchanges latent messages with a “Coprocessor.” It proposes two hypotheses to improve cross-module communication: (H1) augmenting KV-cache communication in a frozen Base and (H2) co-finetuning both models jointly.

**Strengths:**

* **Clear motivation and structure**: The paper clearly frames the problem as evaluating and improving communication between “System-1” (Base) and “System-2” (Coprocessor) reasoning modules.

* **Rigorous experimental baselines and fair token budgeting**: The authors control for latent-token and data budgets and include both continued pretraining and soft-embedding baselines, which exposes when dual architectures add redundant capacity.

* **Comprehensive interpretability analysis**: The cross-capture and silhouette diagnostics offer rare quantitative insight into latent redundancy.

**Weaknesses:**

* **Limited empirical scope and scale**: Experiments are restricted to GPT-2 (124 M) and Qwen-3 (0.6 B) models with modest token budgets. The findings might not generalize to larger LLMs (≥ 7 B parameters) where representational capacity and KV-cache dynamics differ significantly.

* **Unclear evidence for emergent reasoning**: The paper’s conclusions rest on negative results (flat scaling on GSM8K/Countdown), but it lacks task diversity to confirm that the observed redundancy is a general property rather than a dataset artifact.

* **Theoretical contribution remains descriptive**: While the dual-system framing is conceptually appealing, the paper contributes little formal theory or training objectives to explain why latent specialization fails to emerge.

* **Potential training mismatch**: The authors note that pretraining and curriculum finetuning conflict, but do not analyze how or why this mismatch arises, leaving uncertainty about whether the issue is architectural or procedural.

* **Limited methodological advancement**: This paper primarily builds upon an existing dual-module paradigm and explores a few minor variants of it, using empirical experiments to compare their relative strengths and distinctive properties. While such exploration can offer useful insights for future research, the methodological innovation presented here remains limited.

**Questions:**

* **Alternative latent objectives**: Have you tried explicit decorrelation or orthogonality regularizers to encourage latent specialization, or contrastive losses to diversify Coprocessor representations?

* **Ablation of communication depth**: For Hypothesis 1, have you examined partial-layer cache concatenation (e.g., top-k layers only) to see if selective coupling can balance compute and communication efficiency?

* **Cross-task generalization**: Could you include additional reasoning benchmarks (e.g., MATH, SVAMP) to test whether the “redundant latents” phenomenon persists across reasoning domains?

* Can you further elaborate on how the findings presented in this work concretely support or inspire the two promising directions mentioned at the end of the paper?

---

> ### Author Response · Authors · 2025-11-27
>
> Dear Reviewer Pkeg,
>
> We thank the reviewer for praising the clear motivation and structure of the paper regarding the evaluation of System-1 and System-2 modules. We also appreciate the positive feedback on our rigorous experimental baselines and the comprehensive interpretability analysis regarding latent redundancy. We also thank the reviewer for their helpful comments and we have made a response for each of their comments along with suggested changes to the paper. We have addressed some points in a non sequential way if some comments were related to each other.
>
> > Have you tried explicit decorrelation or orthogonality regularizers...?
>
> Yes. We are very grateful for this suggestion, as implementing it revealed a critical mechanism that significantly strengthened our analysis. We added **Section 4.4, pp.9-10** ("Can Explicit Regularization Fix Latent Collapse?") to the paper, detailing the objective:
>
> $\mathcal{L}\_{orth} = \frac{1}{N\_L(N\_L-1)} \sum_{i \neq j} (Z\_i Z\_j^T)^2$
>
> This ablation tested $\mathcal{L}_{orth}$ across our three settings (large-scale pretraining, GSM8K, and Countdown). We invite you to review the novel section (mainly **Figure 5**) in the revised manuscript) for the full experimental details. The notable positive results were to show that on the Countdown task:
>
> - We can force the latents into specialized, non-overlapping subspaces, as visualized in the new **Figure 5C.**
> - Restore monotonic scaling: Unlike the unregularized model, the regularized dual-model effectively utilizes larger latent budgets ($N_L$).
> - Achieve superior robustness: The regularized model gains +11.5% accuracy at $N_L=16$ (62.6% vs 51.1%) compared to the unregularized baseline.
>
> > While the dual-system framing is conceptually appealing, the paper contributes little formal theory or training objectives to explain why latent specialization fails to emerge.
>
> The results from the new orthogonality experiment (**Section 4.4**)—inspired by your previous question—provide the empirical mechanism regarding why this failure occurs. We observed a distinct trade-off: reasoning improved systematically with orthogonality (Countdown, **Fig 5B**), but general modeling perplexity strictly degraded (**Fig 5A**).
>
> To explicitly articulate this mechanism in the paper, we updated **Section 4.4** with the following analysis:
>
> "**This specialization, however, appears antagonistic to large-scale language modeling... increasing $\lambda$ degrades validation perplexity. This suggests that for general language distributions, the model prefers using latents as redundant "bandwidth" to propagate confident predictions (a semantic shortcut) rather than distinct reasoning steps. Consequently, the "System 2" structure required for reasoning might be locally suboptimal for pretraining, creating a tension where the objective naturally favors the "System 1" geometry of redundancy.**"
>
> This clarifies that the lack of specialization is not an architectural failure, but appears to be a direct consequence of the objective mismatch between next-token prediction and structured reasoning.
>
> > Can you further elaborate on how the findings... concretely support or inspire the two promising directions...?"
>
> Our findings directly substantiate the two directions proposed in the conclusion:
> - Direction (i): Designing objectives for diversity:
> The Countdown experiment (**Section 4.4**) demonstrates that when the objective is altered to explicitly reward specialization (orthogonality), the Coprocessor’s reasoning capability improves systematically with $N_L$. This indicates that the barrier to reasoning is partly an objective alignment problem, not solely an architectural one.
> - Direction (ii): Training schedules that shape latent spaces.
> We thank the reviewer for prompting us to re-examine our framing of the curriculum results. We realized our original text lacked clarity and failed to explicitly mention the performance benefits. We have now clarified in **Section 4.3.1** that while the curriculum induces only a "small" geometric shift, it drives a massive performance gain, suggesting the schedule is doing the heavy lifting of latent reorganization: "**While the global silhouette change is slim, it marks a qualitative shift... This geometric reorganization coincides with a substantial performance gain for Hypothesis 2, improving from 11.2% (without curriculum) to 31.5%. This suggests that the curriculum induces an important shift for reasoning even if the specific dynamics of this shift remain to be fully characterized.**"
>
> This highlights that the schedule (curriculum) manages to reshape the latent space into a functional state that simple training struggles to reach, strongly motivating the need for schedules that can gradually bridge this gap.

---

> > ### Author Response · Authors · 2025-11-27
> >
> > > Limited empirical scope and scale: Experiments are restricted to GPT-2 (124 M) and Qwen-3 (0.6 B) models with modest token budgets. The findings might not generalize to larger LLMs (≥ 7 B parameters) where representational capacity and KV-cache dynamics differ significantly.
> >
> > We thank the reviewer for raising the important question regarding the generalizability of our findings to larger models (≥ 7B parameters).
> >
> > We believe that rigorous "debugging" of the architectural mechanism—specifically our interpretability analysis revealing latent collapse and the subsequent fix via regularization—must be established on smaller scales before justifying the massive compute required for larger runs. Scaling up a flawed mechanism (inefficient compute addition) would yield little insight; proving the mechanism works via our stress tests and baselines is the necessary precursor to scaling.
> >
> > By validating our hypotheses on GPT-2 and Qwen-3, we isolate the interaction dynamics between the Base and Coprocessor without the confounding factors of massive pre-training data or emergent capabilities that only appear at scale. Our results suggest that future work should prioritize designing better objectives for latent specialization, which remains an open problem regardless of model size.
> >
> > > Limited methodological advancement: This paper primarily builds upon an existing dual-module paradigm and explores a few minor variants of it, using empirical experiments to compare their relative strengths and distinctive properties. While such exploration can offer useful insights for future research, the methodological innovation presented here remains limited
> >
> > We respectfully argue that while our architectural changes are incremental, our methodological contribution lies in the rigorous deconstruction of the dual-module paradigm itself. Most prior works in latent reasoning optimize for benchmark scores. Our work provides a systematic stress-test and geometric analysis (via latent collapse) to explain why these methods struggle to generalize. By identifying that a failure mode could be objective-misalignment (as shown by our new orthogonality results) rather than just capacity, we provide a correction to the field's trajectory, steering future methodology towards designing objectives/architectures for latent specialization.
> >
> > > The paper’s conclusions rest on negative results (flat scaling on GSM8K/Countdown), but it lacks task diversity to confirm that the observed redundancy is a general property rather than a dataset artifact.
> > And
> > > Cross-task generalization: Could you include additional reasoning benchmarks (e.g., MATH, SVAMP) to test whether the “redundant latents” phenomenon persists across reasoning domains?
> >
> > We thank the reviewer for this constructive suggestion. We agree that substantiating a "negative result"—specifically, that current dual-model objectives lead to latent collapse rather than structured reasoning—requires testing across broader domains.
> >
> > To address this, we have added **Appendix E.3**, which extends our analysis to two additional arithmetic reasoning benchmarks: SVAMP and MultiArith. We excluded the MATH benchmark that the reviewer had proposed as this dataset is too difficult for models of that size.
> >
> > We utilized the Qwen-3 dual-model variant (Hypothesis 2) for these experiments. We did not include GPT-2 for these specific tasks because, unlike Countdown (infinite generation) or GSM8K (where we utilized the large-scale CoT data available from the Coconut paper to build a curriculum), SVAMP and MultiArith have limited training set sizes. We found that without a CoT-based curriculum or massive synthetic data, the smaller GPT-2 model did not learn the task above random baselines.
> >
> > However, the Qwen-3 results on these new tasks strongly corroborate the findings in our main paper:
> > - Latent Scaling Fails to Improve Robustness: As shown in our new **Figure 13 (Appendix E.3)**, increasing the latent budget $N_L$ from 2 to 16 results in flat performance on SVAMP ($\approx 42\%$) and degrading performance on MultiArith (dropping from $\approx 28\%$ to $<20\%$). This mirrors the behavior we observed on GSM8K and Countdown, confirming that simply adding latent capacity does not automatically yield better reasoning.
> > - Latents Exhibit Collapse: Even though the model achieves moderate accuracy (40.6% on SVAMP), our interpretability metrics confirm that the latent space collapses. We observed high off-diagonal variance capture ($\overline{H}_{off} \approx 0.95$ for both tasks) and negative silhouette scores ($-0.011$ for MultiArith, $-0.056$ for SVAMP).

---

> > > ### Author Response · Authors · 2025-11-27
> > >
> > > > Ablation of communication depth: For Hypothesis 1, have you examined partial-layer cache concatenation (e.g., top-k layers only) to see if selective coupling can balance compute and communication efficiency?
> > >
> > > While we did not strictly ablate "top-k" layers for efficiency, we did explore two variations of selective coupling for Hypothesis 1 focused on the middle layers.
> > >
> > > These were motivated by findings that the middle layers of LLMs encode more abstract representations, whereas early and late layers tend to be more "token-aligned" (Lad et al., 2024). We hypothesized that focusing on abstract representations from these Coprocessor layers might help for reasoning. Specifically, we tested:
> > > - Initialization Strategies: We attempted initializing the Coprocessor by cloning only the middle layers of the Base (and duplicating them). The goal was to bias the latent stream towards abstract representations from the start, rather than full-depth emulation.
> > > - Targeted Masking (Hypothesis 1): We experimented with masking the Base's attention so that it attended to the Coprocessor's cache only at these middle layers (effectively a "middle-k" partial concatenation).
> > >
> > > Despite these motivated interventions, we did not observe systematic improvements over the simpler, full-layer cache concatenation baseline. Consequently, we adhered to the simpler full-layer configuration to prioritize a rigorous evaluation of the core architecture.
> > >
> > > **Reference**:
> > >
> > > Vedang Lad, Wes Gurnee, and Max Tegmark. The Remarkable Robustness of LLMs: Stages of Inference?. NeurIPS, 2025.

---

### Official Review · Reviewer_2Jz5 · 2025-10-29

**Soundness:** 3
**Presentation:** 2
**Contribution:** 2
**Rating:** 4
**Confidence:** 3

**Summary:**

The paper tests dual-model latent reasoning where a Base LLM exchanges “thought” tokens with a Coprocessor. Of two upgrades over Liu et al. (2024), H2 (joint finetuning) beats H1 (KV-cache concat with frozen Base)—but a single-model soft-embedding baseline with the same latent budget nearly matches H2 and surpasses H1, suggesting the dual setup mostly adds compute, not better reasoning.

**Strengths:**

1. The paper poses clear, falsifiable hypotheses about cross-module communication in dual-model latent reasoning.

2. The evaluation in the paper matches latent budgets across methods and adds a strong single-model soft-embedding baseline for fair comparison.

3. The proposed three-pass training protocol reduces shortcutting and clarifies where gains originate.

**Weaknesses:**

1. The paper’s motivation feels underdeveloped. The work is framed as improving the KV-Coprocessor within its own paradigm, but the generality and adoption of KV-Coprocessor remain unsettled, making this study read more like a targeted extension of that framework than a standalone contribution.

2. The dual-model setups introduce additional trainable components relative to the soft-embedding baseline, leaving open whether observed gains stem from architectural choices or simply extra capacity/compute.

3. The ablation of the each component of the proposed structure is missing in the current work.

4. The KV-concat pathway (H1) may be sensitive to sequence length or attention scaling, yet length-generalization tests are absent, leaving robustness claims underexplored.

**Questions:**

See the weaknesses part.

---

> ### Author Response · Authors · 2025-11-27
>
> Dear Reviewer  2Jz5,
>
> We thank the reviewer for highlighting that our paper poses clear, falsifiable hypotheses about cross-module communication. We also appreciate the recognition of our rigorous evaluation. We also thank the reviewer for their helpful comments and we have made a response for each of their comments along with suggested changes to the paper.
>
> > The paper’s motivation feels underdeveloped. The work is framed as improving the KV-Coprocessor within its own paradigm, but the generality and adoption of KV-Coprocessor remain unsettled, making this study read more like a targeted extension of that framework than a standalone contribution.
>
> We thank the reviewer for this perspective. Fundamentally, we believe there is strong motivation—grounded in dual-process theories and supporting neuroscience evidence regarding distinct substrates—to explore architectures that disentangle reasoning from fluent text generation. We hypothesize that these distinct cognitive processes should not be forced to share the same representational space. While we aimed to clearly articulate this motivation in the introduction, we would be happy to expand on any specific aspects that the reviewer felt were under-explained.
>
> Regarding the scope, we view this work not only as a targeted extension of one specific architecture, but also as a diagnostic of this broader dual-system hypothesis. We selected the KV-Coprocessor as our testbed because it offers the current cleanest architectural separation between the two systems, making it the obvious choice for a first step into studying latent communication.
>
> Our contribution extends beyond optimizing a single model in two ways:
>
> - Diagnosis: We identify why these architectures often fail in practice: without explicit constraints, they default to inefficient compute allocation (latent collapse) rather than true reasoning.
> - Solution: With the new results added in this revision (**Section 4.4**), we demonstrate that enforcing latent specialization can, in some scenarios, prevent this collapse and restore performance scaling with respect to the latent budget.
>
> We believe this offers critical guidance for future research on latent communication, specifically highlighting the need to couple modular architectures with objectives that explicitly enforce geometric structure to enable robust reasoning.
>
> > The dual-model setups introduce additional trainable components relative to the soft-embedding baseline, leaving open whether observed gains stem from architectural choices or simply extra capacity/compute.
>
> We completely agree that distinguishing between architectural benefits and simple parameter scaling is critical. To strictly verify this, we have added a new experiment in **Appendix F (Figure 12)** specifically for this revision:
>
> **" Setup: We train a ‘’small to medium'' GPT-2 with 16 layers, 16 heads, and 1024-dimensional embeddings, resulting in approximately 254M parameters. This matches the aggregate parameter count of our dual-system (124M Base + 124M Coprocessor). We first pre-train this model from scratch using the NanoGPT framework (Karpathy et al., 2022) to serve as a parameter-matched baseline. We note that we limited this ablation to the GPT-2 family, as performing full pre-training of a parameter-matched Qwen model ($\approx$ 1.2B parameters) from scratch is unrealistic for the scope of this paper."**
>
> The results on GPT-2 were decisive: The single parameter-matched model attained lower perplexity than the Dual-Model system.
>
> This directly answers the open question: the gains observed in the Dual-Model were indeed due to "extra capacity," but the Dual architecture is an inefficient way to deploy that capacity compared to a unified model of the same size. This reinforces our paper's primary conclusion that current dual-model designs add compute without unlocking qualitatively better reasoning, unless specific regularizations (like our new $\mathcal{L}_{orth}$) are applied (new **Section 4.4**).
>
> **References:**
>
> Andrej Karpathy. nanogpt. https://github.com/karpathy/nanoGPT, 2022. The simplest,fastest repository for training/finetuning medium-sized GPTs.

---

> > ### Author Response · Authors · 2025-11-27
> >
> > > The ablation of the each component of the proposed structure is missing in the current work.
> >
> > We appreciate the reviewer pointing this out. We strive to be as exhaustive as possible; however, as noted in Section 4.1, our computational budget is constrained (each run requires approx. 3 days on 8xA100s), which limits the feasible search space for a full combinatorial ablation. Nevertheless, we agree that understanding the sensitivity of the proposed components is improved by further data. We have added a new section, **Appendix G: Extra Ablations**, to the revised paper. Specifically:
> >
> > - Latent Budget ($N_L$) for Hypothesis 1: We provide perplexity curves ablating $N_L$ for the frozen-base/KV-concatenation model (**Figure 13**), complementing the analysis of Hypothesis 2 in the main text.
> > - Ahead Tokens ($N_A$): Following the design of Liu et al. (2024b), we conducted ablations on the number of "ahead tokens" used for supervision (Table 6). We find that while $N_A=16$ (our default) is robust, slight gains can be found at $N_A=32$ for GPT-2, though $N_A=16$ remains optimal for Qwen-3. We focus these ablations on arc_easy, which is standard for gauging reasoning capabilities in models of this size (GPT-2 124M / Qwen-3 0.6B), observing trends consistent with our main reasoning benchmarks.
> > - Finally we perform some length generalization ablations which we expand on in the next response.
> >
> > > The KV-concat pathway (H1) may be sensitive to sequence length or attention scaling, yet length-generalization tests are absent, leaving robustness claims underexplored.
> >
> > We thank the reviewer for their feedback and we agree that the potential sensitivity of the KV-concatenation mechanism to sequence length deviations is indeed a critical robustness check.
> >
> > We have added a new section, **Appendix G.1**: Length Generalization, where we evaluate the perplexity of our trained models on sequences extending beyond their training context length $S$. We tested generalization up to $2\times$ the training length ($S \times \{1.0, 1.125, 1.25, 1.5, 2.0\}$).
> >
> > For the GPT-2 experiments, we explicitly set the training sequence length to $S=512$ (half of its original pretraining context of 1024). This ensures that our extrapolation tests (up to $2\times S = 1024$) measure the architectural generalization capabilities of our method without being confounded by the hard positional limits of the underlying pre-trained model.
> >
> > Our findings shown in **Figure 14** are described as follows in the paper:
> > - "**Hypothesis 1 (Frozen-Base KV Augmentation): Contrary to concerns that cache concatenation might disrupt attention mechanisms at longer lengths, Hypothesis 1 demonstrates strong robustness. For Qwen-3, performance remains stable across all tested lengths. For GPT-2, the model generalizes well up to $1.5\times$ length, with only slight degradation observed at $2\times$ length.**
> > - **Hypothesis 2 (Co-finetuned):} The fully finetuned dual-model system shows architecture-dependent generalization. On Qwen-3 (which uses Rotary Positional Embeddings), H2 generalizes effectively. However, on GPT-2 (which uses absolute positional embeddings), H2 fails to generalize, exhibiting rapid perplexity degradation when inputs exceed the training context.**
> >
> > **This suggests that these dual-model architectures are robust to length generalization provided they utilize modern relative positional embeddings (like RoPE in Qwen-3), whereas architectures with absolute positional embeddings (as in GPT-2) may struggle to generalize in the co-finetuned setting.**"

---

### Official Review · Reviewer_pnWg · 2025-10-31

**Soundness:** 3
**Presentation:** 2
**Contribution:** 2
**Rating:** 2
**Confidence:** 3

**Summary:**

The paper studies dual-architecture latent reasoning, which contains a base model and a coprocessor. The authors tested two hypotheses aimed at improving latent communication compared to a previous work [1]. The evaluation of the two hypotheses suggests that the current dual designs mostly add computation rather than improving reasoning, which is supported by the marginal gain on GSM8k, ProsQA, and the countdown stress test dataset by scaling the latent-token budget beyond small values, and overlapping spaces in latent analysis.

**References:**

[1] Liu, Luyang, Jonas Pfeiffer, Jiaxing Wu, Jun Xie, and Arthur Szlam. "Deliberation in latent space via differentiable cache augmentation." arXiv preprint arXiv:2412.17747 (2024).

**Strengths:**

1. The paper studies an important and interesting question: whether the dual model really benefits reasoning by effective communication, or whether it merely adds computation (even in an inefficient way)?

2. The paper conducts a more careful analysis of a previous work, which uncovers interesting results, such as using a single model can achieve similar performance gain as the dual system.

**Weaknesses:**

1. The major concern is that the scope of this paper might be a bit narrow. The conclusion that the current dual system for latent reasoning is not efficient is based on the analysis of one previous work, which makes the conclusion not fully convincing or solid. The contribution is also limited, since the paper did not propose effective methods to mitigate it.

2. The definition of the whole process can be more precise. Many refer to Liu et al. (2024b), but it would be more helpful for readers unfamiliar with the work to provide the mathematical formulations in greater detail.

3. The flow of the paper could be improved for reading. Some sentences are not very coherent within the context, and some seem broken.

4. “A single forward pass could allow the Base model to shortcut … defeating the purpose of the technique.” Although this might make sense, I think there should be some evidence, even if it is indirect.

**Questions:**

1. What does the ahead token N_A mean? Is it defined anywhere in the paper? For the latent augmentation, what does M refer to (i.e., what is the difference between multiple augmentations for the same sequence)

2. On page 6, the authors mentioned, “A single LLM with the same aggregate parameter count would almost certainly do better.” But the result for Qwen3 benchmarks shows “Dual models fare better: Hyp. 1 +6.5 pp (47.2 vs. 40.7%); Hyp. 2 +8.7 pp (49.4%).”. Am I missing anything?

3. In diagnostic 1 (line 397), what does $n\_i$ mean?

4. For diagnostic 1, I wonder what the typical dimension of $P\_i$ is. If $P\_i$ is close to the identity matrix, then $H\_{ij}$ should always be very high.

---

> ### Author Response · Authors · 2025-11-27
>
> Dear Reviewer pnWg,
>
> We thank the reviewer for recognizing that our paper studies an important and interesting question regarding the true benefits of dual-model latent reasoning. We also appreciate the acknowledgment of our careful analysis of previous work and the interesting results uncovered. We also thank the reviewer for their helpful comments and we have made a response for each of their comments along with suggested changes to the paper.
>
> > The major concern is that the scope of this paper might be a bit narrow. The conclusion that the current dual system for latent reasoning is not efficient is based on the analysis of one previous work, which makes the conclusion not fully convincing or solid. The contribution is also limited, since the paper did not propose effective methods to mitigate it.
>
> We thank the reviewer for their assessment. To directly address the concern that the contribution is limited by a lack of "effective methods to mitigate" these issues, we have significantly expanded the paper (new **Section 4.4, pp. 9-10**) to propose and validate a solution.
>
> Since our initial analysis diagnosed that the dual-model inefficiency stemmed from latent representations collapsing into redundant subspaces, we introduced an explicit regularization to penalize similarity between latents. This new mitigation strategy yielded substantial improvements on the combinatorial Countdown task:
>
> - Forced Specialization: As visualized in the new **Figure 5C**, regularization forces latents into specialized, non-overlapping subspaces, resolving the collapse we originally identified.
> - Restored Scaling: Unlike the unregularized baseline, the regularized dual-model effectively utilizes additional latent capacity, restoring monotonic performance scaling as $N_L$ increases.
> - Superior Robustness: This approach achieved a +11.5% accuracy gain at $N_L=16$ (62.6% vs 51.1%) compared to the unregularized baseline.
>
> These new results shift the contribution from a purely negative diagnosis to a constructive demonstration of how dual-system latent reasoning can be engineered to succeed. This directly speaks to the concern regarding the scope: we clarify that we utilize this specific dual-model architecture not merely to analyze a single prior work, but as a controlled testbed to diagnose the fundamental bottlenecks of latent communication for reasoning language models.

---

> > ### Author Response · Authors · 2025-11-27
> >
> > > “A single forward pass could allow the Base model to shortcut … defeating the purpose of the technique.” Although this might make sense, I think there should be some evidence, even if it is indirect.
> >
> > We thank the reviewer for pressing us on this point. We agree that this claim relies on an optimization hypothesis, and we have revised the text to cite the relevant literature.
> >
> > Our reasoning is two-fold:
> >
> > 1. Optimization Shortcut (Path of Least Resistance):
> >
> > In a single-pass implementation, the optimizer is strongly incentivized to ignore the noisy latent channel (the randomly initialized Coprocessor) and rely entirely on the Base model’s internal representations. This mirrors Shortcut Learning (Geirhos et al., 2020), where models exploit easier surface features over complex intended mechanisms, and VAE Posterior Collapse (Bowman et al., 2016), where strong decoders ignore noisy latent variables entirely to minimize loss via the path of least resistance.
> >
> > 2. Engineering Feasibility:
> >
> > More importantly, it is not clear how to implement a genuine single forward pass for this architecture in standard frameworks and we could not get a response from the authors of that paper on guidance for the single forward pass. Since the Coprocessor ($C_\phi$) is a distinct model that must process the Base model's output ($KV_\theta$) before the Base model can resume decoding, there is a strict sequential dependency. "Single pass" implies a fused computational graph that handles two distinct parameter sets interleaved dynamically.
> >
> > Therefore, we have revised **Section 4.1** to clarify that our Three-Pass loop is a deliberate choice to resolve both this engineering ambiguity and the optimization risk:
> >
> > "**However, we hypothesize that a single forward pass creates an optimization shortcut: since the Base model is pre-trained and competent while the Coprocessor begins as initialized noise, the joint system is incentivized to ignore the noisy latent channel in favor of the Base's internal representations. This reflects the ideas around path of least resistance in deep learning (Bowman et al., 2015; Geirhos et al., 2020). Furthermore, it is technically unclear how to implement such a bidirectional dependency (Base -> Coprocessor -> Base) within a single forward pass, and (Liu et al., 2024) do not provide implementation details to clarify this. To ensure the Base strictly conditions on the Coprocessor's output and to avoid optimization collapse, we adopt a strict three-pass loop (Figure 6) that architecturally enforces the dependency.**"
> >
> > **References**:
> >
> > Bowman, S. R., et al. (2016). Generating sentences from a continuous space.
> > Geirhos, R., et al. (2020). Shortcut learning in deep neural networks.

---

> ### Author Response · Authors · 2025-11-27
>
> > The definition of the whole process can be more precise. Many refer to Liu et al. (2024b), but it would be more helpful for readers unfamiliar with the work to provide the mathematical formulations in greater detail.
>
> AND
> > The flow of the paper could be improved for reading. Some sentences are not very coherent within the context, and some seem broken.
>
> AND
> > What does the ahead token N_A mean? Is it defined anywhere in the paper? For the latent augmentation, what does M refer to (i.e., what is the difference between multiple augmentations for the same sequence)
>
> We have grouped these three comments as they all rightly identify a need for greater formal precision and better definitions of our variables. We agree that the initial draft relied too heavily on familiarity with Liu et al. (2024b), which impacted the clarity and flow for a broader audience.
>
> To address this, we have significantly rewritten the 'Problem Setting' (**Section 3.1**) and added a detailed formulation (**Appendix B.1**) to rigorously define the process.
>
> **1.** Clarifying $M$ and $N_A$:
>
> We now explicitly define these variables immediately upon introduction. To answer your specific questions: $M$ refers to the number of augmentation sites (which allows us to train on multiple segments per sequence in parallel), and $N_A$ refers to the "ahead" tokens (the specific window of future tokens used to calculate the loss).
>
> Implemented text in **Section 3.1**:
>
> **"We define a set of $M$ augmentation sites indices $\mathcal{T}=\{t_1, \dots, t_M\}$ within the sequence $x$. At each site $t_m$, a Coprocessor $C_{\phi}$ with parameters $\phi$ reads $KV_{\theta}(x)$ up to $t_m$ together with $N_{L}$ learnable soft tokens... During training, gradients flow from the next $N_A$ 'ahead' tokens (the supervision window) back to the latents."**
>
> **2.** Detailed Mathematical Formulation (**Appendix B.1**):
>
> To ensure the "whole process" is precise and to improve the technical flow, we added the explicit loss formulation showing exactly how $M$ and $N_A$ interact during the decoding phase.
>
> Implemented formulation in **Appendix B.1**:
>
> "**Here we formalize the three-pass process described in Section 3.1**:
>
> **Let $x = (x_1, \dots, x_S)$ be a sequence. We select a set $\mathcal{T} = \{t_1, \dots, t_M\}$ of $M$ augmentation indices, sampled uniformly from valid sequence positions.**
>
> **(i) KV-cache generation: We compute the frozen Key-Value cache for the entire sequence once: $KV_{\theta}(x) = B_{\theta}(x)$.**
>
> **(ii) Latent augmentation: For each augmentation site $t_m \in \mathcal{T}$, the Coprocessor generates a sequence of latents $Z^{(m)}$. It attends to the Base cache prefix $KV_{\theta}(x_{<t_m})$ and $N_L$ learnable soft tokens $S_{soft}$:
> $$Z^{(m)} = C_{\phi}(KV_{\theta}(x_{<t_m}) \oplus S_{soft}) \in \mathbb{R}^{N_L \times d}$$**
>
> **(iii) Decoding: The Base model predicts the next $N_A$ tokens (the "ahead" tokens) conditioned on the injected latents. The function $\text{inject}(\cdot)$ abstracts the specific interaction mechanism (embedding injection for Hyp. 2 or cache concatenation for Hyp. 1). The loss is computed only over these $N_A$ tokens for every augmentation site $m$, maximizing the likelihood:**
> **$$\mathcal{L} = \sum_{m=1}^{M} \sum_{j=1}^{N_A} \log p_{\theta}\left(x_{t_m+j} \mid \text{inject}\left(KV_{\theta}(x_{<t_m}), Z^{(m)}\right), x_{t_m : t_m+j-1}\right)$$**
> **$M$ allows us to train on multiple randomly sampled reasoning segments per sequence in parallel, while $N_A$ defines the window of future tokens that provide the supervisory signal.
> "**
>
> We believe these revisions clarify the definitions and strictly define the mathematical process. If any specific part of the flow remains unclear, we are more than happy to make further adjustments.

---

> ### Author Response · Authors · 2025-11-27
>
> > On page 6, the authors mentioned, “A single LLM with the same aggregate parameter count would almost certainly do better.” But the result for Qwen3 benchmarks shows “Dual models fare better: Hyp. 1 +6.5 pp (47.2 vs. 40.7%); Hyp. 2 +8.7 pp (49.4%).”. Am I missing anything?
>
> We thank the reviewer for this careful observation. They are correct that on the Qwen-3 benchmarks, the Dual models (Hyp 1 & 2) outperform the "Soft Embedding" baseline. The confusion likely stems from the parameter count of the baselines used in Table 1 versus the "aggregate parameter count" mentioned in the text.
>
> To clarify:
>
> - The "Soft Embedding" baseline in Table 1 uses a single model instance (e.g., 0.6B parameters for Qwen-3).
> - The Dual Models (Hyp 1 & 2) utilize both a Base and a Coprocessor, effectively doubling the active parameters during the forward passes (e.g., ~1.2B parameters total for Qwen-3).
>
> When we stated that "A single LLM with the same aggregate parameter count would almost certainly do better," we meant that a single 1.2B parameter model would likely outperform the 0.6B + 0.6B Dual setup.
>
> To make this distinction clearer and validate our claim, we have added a new experiment in **Appendix F (Figure 12)** specifically for this revision.
>
> In this new analysis, we trained a single larger GPT-2 model (250M parameters) to match the aggregate size of the dual-model setup (124M Base + 124M Coprocessor). The results confirmed our hypothesis: the single parameter-matched model achieved lower perplexity than the dual-model architecture.
>
> Thus, our conclusion remains that while the Dual architecture beats the smaller single baseline on Qwen-3, it is doing so primarily by adding compute/parameters, and it is doing so inefficiently compared to simply scaling up a single dense model.
>
> > In diagnostic 1 (line 397), what does n_i  mean?
>
> We thank the reviewer for pointing out this unclear notation. In the original draft, $n_i$ was intended to denote the number of evaluation samples.
>
> To eliminate any ambiguity, we have replaced $n_i$ with $N_{eval}$ in the revision and explicitly defined it in Section 4.3. The updated text (marked in red) now reads:
>
> **"Let $X_{i} \in \mathbb{R}^{N_{eval} \times d}$ be the row-centered activations for latent $i$ collected across the full evaluation set (where $N_{eval}$ is the number of task-evaluation examples)."**
>
> > For diagnostic 1, I wonder what the typical dimension of P_i is. If P_i is close to the identity matrix, then $H_{ij}$  should always be very high.
>
> We strongly agree with the reviewer that checking the dimension of the subspace $P_i$ is crucial for interpreting Diagnostic 1. To address this, we have added **Appendix E.1** (Effective Subspace Dimensions) to explicitly quantify the dimensionality of the PCA subspaces used in our metric (which explain 97% of the variance).
>
> The table confirm that the subspaces are effectively low-rank relative to the model dimension ( $d=768$):
>
> - Large Scale Training: Average rank $\approx 2$ (0.2 % of $d$).
> - Countdown: Average rank $\approx 60$ (7.8 % of $d$).
> - GSM8K: Average rank $\approx 161$ (21 % of $d$).
>
> In all cases, the subspace dimension is significantly lower than the ambient dimension ($rank \ll 768$). This rules out the hypothesis that $P_i$ is close to the identity matrix and confirms that the high overlap values we report represent a genuine geometric collapse of the latent representations.

---

### Official Review · Reviewer_x3BX · 2025-11-01

**Soundness:** 2
**Presentation:** 1
**Contribution:** 2
**Rating:** 6
**Confidence:** 2

**Summary:**

This paper examines dual-architecture latent reasoning where a Base model exchanges latent messages with a Coprocessor to improve reasoning capabilities. The authors test two hypotheses to strengthen communication over existing work - increasing channel capacity via cache augmentation and learning communication through joint finetuning - but find that a simpler soft-embedding baseline nearly matches the best dual-model variant, suggesting the approach mostly adds compute rather than enabling qualitatively better reasoning.

**Strengths:**

The paper provides a thorough empirical evaluation across multiple model sizes (GPT-2, Qwen-3) and tasks with proper baselines, which is refreshing. The interpretability analysis using cross-capture heatmaps and silhouette scores to examine latent specialization is insightful and reveals that latents tend to occupy overlapping subspaces rather than specializing. The three-pass training implementation cleanly separates latent computation from next-token prediction, avoiding potential shortcuts.

**Weaknesses:**

The wrting is bad.
The negative results, while honestly reported, limit the impact .. the dual-model architecture doesn't deliver on its promise of System-2 reasoning. The soft-embedding baseline performs nearly as well with half the parameters, which undermines the main architectural contribution. The experiments are limited to smaller models due to compute constraints, and it's unclear if findings generalize to larger scales where the original work showed stronger results.

**Questions:**

Have you tried other communication mechanisms beyond cache augmentation, like attention-based message passing between modules?

---

> ### Author Response · Authors · 2025-11-27
> **1/2**
>
> Dear reviewer x3BX,
>
> We thank the reviewer for finding our empirical evaluation thorough and refreshing, and for highlighting the insightfulness of our interpretability analysis regarding latent specialization. We also thank the reviewer for their helpful comments and we have made a response for each of their comments along with suggested changes to the paper.
>
> > The negative results, while honestly reported, limit the impact .. the dual-model architecture doesn't deliver on its promise of System-2 reasoning. The soft-embedding baseline performs nearly as well with half the parameters, which undermines the main architectural contribution.
>
> We thank the reviewer for this candid assessment. We agree that while negative results are scientifically vital—preventing the community from pursuing dead-ends and clarifying why certain architectures fail despite their intuitive appeal—coupling them with a constructive path forward significantly strengthens the contribution.
>
> To address your concern that the architecture "*doesn't deliver on its promise of System-2 reasoning*," we have significantly expanded our analysis in the updated paper to include positive evidence (**Section 4.4, pp. 9-10**) that addresses this limitation. Specifically, by introducing an explicit orthogonality regularization ($\mathcal{L}_{orth}$), we were able to show on the Countdown task that:
>
> - We can force the latents into specialized, non-overlapping subspaces, as visualized in the new Figure 5C.
> - Restore monotonic scaling: Unlike the unregularized model, the regularized dual-model effectively utilizes larger latent budgets ($N_L$).
> - Achieve superior robustness: The regularized model gains +11.5% accuracy at $N_L=16$ (62.6% vs 51.1%) compared to the unregularized baseline.
>
> These new results provide the evidence requested: the dual-model architecture seems to be able to deliver System-2 reasoning in some scenarios and outperform simpler baselines, provided the training objective explicitly enforces the geometric structure required for planning. We believe this provides actionable insight for future research on latent reasoning.
>
> > The experiments are limited to smaller models due to compute constraints, and it's unclear if findings generalize to larger scales where the original work showed stronger results.
>
> We acknowledge that our experiments focus on smaller models (GPT-2 and Qwen-3) due to compute constraints. However, we respectfully disagree with the premise that the original work (Liu et al., 2024) demonstrated "stronger results." In fact, our analysis suggests the opposite:
>
> - Stronger Marginal Gains: When following the original paper's protocol (comparing the dual-model against the initial Base Checkpoint), our replication actually yields larger gains than reported in the original work. As detailed in Section 4.1.2, we observe perplexity drops of 2.0 (GPT-2) and 2.5 (Qwen-3) —an order of magnitude larger than the reductions reported by Liu et al. on Gemma-2B (typically <0.3 for 16 latents).
> - The Baseline Illusion: The original work appeared to show "strong" results because they compared the trained dual-model against a frozen base checkpoint—effectively comparing against a single model with half the aggregate parameters and significantly less training.
>
> Therefore, our findings do not indicate that the method fails because of the smaller scale; rather, they reveal that the gains reported in the original work were likely due to the additional training tokens rather than the architectural innovation itself. We believe that rigorous "debugging" of the architectural mechanism—specifically our interpretability analysis revealing latent collapse and the subsequent fix via regularization —must be established on smaller scales before justifying the massive compute required for larger runs. Scaling up a flawed mechanism (inefficient compute addition) would yield little insight; proving the mechanism works via our stress tests and baselines is the necessary precursor to scaling.

---

> > ### Author Response · Authors · 2025-11-27
> > **2/2**
> >
> > > Have you tried other communication mechanisms beyond cache augmentation, like attention-based message passing between modules?
> >
> > Unfortunately, we did not explore attention-based message passing (such as cross-attention layers) in this study. We made a deliberate choice to restrict our scope to KV-cache-based communication—specifically comparing embedding injection vs. our proposed cache concatenation. Our goal was to isolate the variables within this specific architectural family to understand if latent cache streams alone could support System-2 reasoning.
> >
> > However, within this cache-based paradigm, we did explore several advanced variations motivated by the finding that middle layers in LLMs often encode more abstract representations compared to the token-aligned early/late layers (Lad et al., 2024). Specifically:
> >
> > - Initialization strategies: We attempted initializing the Coprocessor by cloning only the middle layers of the Base (and duplicating them) to bias the latent stream towards abstract representations from the start.
> > - Targeted Masking (Hypothesis 1): We experimented with masking the Base's attention so that it attends to the Coprocessor's cache only at the middle layers. The intuition was to prevent the Base from constraining the Coprocessor's cache to be close to "token space" (as often required by early/late layers), thereby giving the latents more freedom to maintain abstract states.
> >
> > Despite these motivated interventions, we did not observe systematic improvements over the simpler baselines.
> >
> > **References:**
> >
> > Vedang Lad, Wes Gurnee, and Max Tegmark. The Remarkable Robustness of LLMs: Stages of Inference?. NeurIPS, 2025.

---

### Author Response · Authors · 2025-11-27
**Summary of the rebuttal**

We would like to thank all reviewers for their valuable and thoughtful feedback.

- Reviewer x3BX found the “empirical evaluation [...] thorough” and the “interpretability analysis [...] insightful.”
- Reviewer pnWg said the paper “studies an important and interesting question” and that the “careful analysis [...] uncovers interesting results.”
- Reviewer 2Jz5 noted that the work “poses clear, falsifiable hypotheses” and praised the addition of a “strong single-model soft-embedding baseline for fair comparison.”
- Reviewer Pkeg highlighted the “clear motivation” and “rigorous experimental baselines,” calling the “interpretability analysis” comprehensive.

Furthermore, the paper received ratings of ‘good’ on Soundness from 3 reviewers.

However, all reviewers also raised important points and provided helpful suggestions. We were able to incorporate all of these suggestions and believe that doing so has improved our paper significantly. To summarize, we have made the following modifications:

- We introduced an auxiliary loss to penalize latent similarity, which successfully restores performance scaling on the Countdown task; this demonstrates the architecture's potential for true latent reasoning and offers a clear path for future research (see new **Section 4.4, pp. 9-10**).
- We extended our evaluation to two additional reasoning datasets to confirm our findings generalize across tasks (see new **Appendix E.3, p. 20**).
- We added a rigorous parameter-matched baseline to isolate the true architectural benefits from added compute (see new **Appendix F, p. 22**).
- We performed extensive new ablations, including tests on length generalization and hyperparameter sensitivity (see new **Appendix G, p. 23**).
- We clarified the substantial performance gains driven by the training curriculum (see updated **Section 3.1, p. 8** ).


We describe these changes in detail in our responses to the individual reviews below. Please note that we have highlighted key changes in **bold** within our responses, and all revisions in the updated manuscript are highlighted in red. We again want to thank the reviewers for their time and for actively taking part in the review process.

---

> ### Author Response · Authors · 2025-12-02
> **Note to Area Chair: Addressing the consensus critique to justify reconsideration due to cancellation of reviewer engagement**
>
> Dear Area Chair,
>
> We provided an exhaustive rebuttal answering most reviewer requests, but we want to highlight one critical update that addresses the consensus concern shared by all reviewers: that despite the paper's clear motivation and thorough experiments, its impact was limited by "negative results" (though we maintain there is significant value in systematically diagnosing the behavioral and representational failures of these systems).
>
> We are confident that our major revision—specifically the new **Section 4.4, pp. 9-10**—would have led at least three reviewers (the two 4s and the 2) to likely raise their scores, resulting in a **worst-case revised average of 5.5**.
>
> The paper no longer just diagnoses a failure mode, but now validates a concrete path to to make latent communication work, restoring reasoning signatures in a combinatorial task:
> - **Restored Performance Scaling**: The model can now effectively utilize larger latent budgets
> - **Significant Gains**: +11.5 % on the countdown task
>
> We kindly ask you to evaluate the revised manuscript with this shift in mind.
>
> Best regards,
>
> The Authors

---

### Meta-Review · Area_Chair_86G3 · 2026-01-04

**Summary:**

The overall ratings gave by the reviewers are relatively low, with one reject (2), two marginally below (4), and one marginally above (6).

Reviewers are generally concerned with the narrow scope of the paper.

Reviewer x3BX: “… it’s unclear if findings generalize to larger scales ...”

Reviewer Pkeg: “The findings might not generalize to larger LLMs …”

Reviewer pnWg: “Conclusion is based on the analysis of a single previous work.”

Reviewers also have concerns about the limited impact of the paper.

Reviewer x3BX had concerns about the negative results and the architectural contribution.

Reviewer pnWg: “… The contribution is also limited, since the paper did not propose effective methods to mitigate it.”

Reviewer 2Jz5: “… making this study read more like a targeted extension of that [KV-Coprocessor] framework than a standalone contribution.”

Reviewer Pkeg pointed out the theoretical contribution remains descriptive and there is limited methodological advancements.

Some reviewers have concerns about the insufficient analysis and ablation:

Reviewr 2Jz5: lack of analysis on the gains, lack of ablation of each component, lack of length-generalization tests.

Reviewer Pkeg: “… lack task diversity to confirm that the observed redundancy is a general property rather than a dataset artifact”, “… but do not analyze how or why this mismatch arises, leaving uncertainty about whether the issue is architectural or procedural.”

Some reviewers are concerned about the poor writing / presentation.

Reviewer x3BX gave 1 (poor) on the Presentation score, and pointed out “the writing is bad.”

Reviewer pnWg pointed out “the definition of the whole process can be more precise”, and suggested “the flow of the paper could be improved for easing.”



Although the authors expanded the paper, added new experiments, to propose and validate a solution during the rebuttal period, the revisions are significant and it is difficult to judge the quality and soundness of the additional content without extra round of reviews.

**Reviewer Concerns:**

The authors have added more experiments to respond to reviewers’ concerns about narrow scope, limited impact, and insufficient analysis/ablation.

To address the concerns about poor writing / presentation, the authors have rewritten some part of the paper.

**Reviewer Scores:**

Reviewer pnWg might increase their scores from 2 to 3 or 4, but unlikely to give 5 or above.

Reviewers 2Jz5 and Pkeg currently gave 4, and they may keep or increase their current scores to 5.

Reviewer x3BX currently gave 6 and is unlikely to change their overall rating.

---

### Decision · Program_Chairs · 2026-01-26

Reject